# Multiresolution Analysis and Statistical Thresholding on Dynamic Networks

**Raphaël Romero**
Ghent University
`raphael.romero@ugent.be`

**Tijl De Bie**
Ghent University
`tijl.debie@ugent.be`

**Nick Heard**
Imperial College London
`n.heard@imperial.ac.uk`

**Alexander Modell**
Imperial College London
`a.modell@imperial.ac.uk`

## Abstract

Detecting structural change in dynamic network data has wide-ranging applications. Existing approaches typically divide the data into time bins, extract network features within each bin, and then compare these features over time. This introduces an inherent tradeoff between temporal resolution and statistical stability of the extracted features. Despite this tradeoff, reminiscent of time–frequency tradeoffs in signal processing, most methods rely on a *fixed temporal resolution*. Choosing an appropriate resolution parameter is typically difficult, and can be especially problematic in domains like cybersecurity, where anomalous behavior may emerge at multiple time scales. We address this challenge by proposing ANIE (**A**daptive **N**etwork **I**ntensity **E**stimation), a multi-resolution framework designed to automatically identify the time scales at which network structure evolves, enabling the joint detection of both rapid and gradual changes. Modeling interactions as Poisson processes, our method proceeds in two steps: (1) estimating a low-dimensional subspace of node behavior, and (2) deriving a set of novel *empirical affinity coefficients* that quantify change in interaction intensity between latent factors and support statistical testing for structural change across time scales. We provide theoretical guarantees for subspace estimation and the asymptotic behavior of the affinity coefficients, enabling model-based change detection. Experiments on synthetic networks show that ANIE adapts to the appropriate time resolution, and is able to capture sharp structural changes while remaining robust to noise. Furthermore, applications to real-world data showcase the practical benefits of ANIE 's multiresolution approach to detecting structural change over fixed resolution methods. An open-source implementation of the method is available at `https://github.com/aida-ugent/anie`.

## 1 Introduction

Understanding dynamic networks, namely datasets taking the form of sequences of interaction events $(u, v, t)$ between nodes $u$ and $v$ at timestamp $t$ has wide-ranging applications in domains such as contact tracing[16], cybersecurity[34] and urban mobility studies [3, 21]. Despite this domain diversity, temporal networks commonly exhibit two fundamental types of structure: **cross-sectional structure**, where the network is seen as a graph evolving over time, and **longitudinal structure**, where the data at its finest resolution is best modeled as a collection of point processes [33, 30, 29].

At its core, change detection in such networks involves understanding how these two types of structure interact. However, doing so involves an inherent tradeoff. On one hand, identifying cross-

39th Conference on Neural Information Processing Systems (NeurIPS 2025).

sectional structure-such as communities-requires aggregating events over a sufficiently wide time window to achieve statistical stability. On the other hand, imposing a certain resolution of analysis may obscure transient events which occur at higher temporal resolutions. This mirrors the time–frequency tradeoff in signal processing, where narrow time windows reveal high-frequency details but miss low-frequency trends, and wide windows improve frequency resolution at the cost of temporal localization.

In practice, the choice of an appropriate time resolution is a challenge which manifests in a variety of ways, such as selecting the number of timesteps at which to evaluate dynamic node embeddings [41, 42], or selecting a bandwidth in order to derive smooth temporal signals from the dynamic network [29]. Often this challenge is resolved by selecting a resolution which seems to correspond to some characteristic period or frequency derived a priori from domain knowledge [17, 20, 29]. However in applications such as cybersecurity [33, 35, 13], where time-localization of anomalous event is essential, or more broadly community detection [43], where node behaviors may align at different resolution levels, such an arbitrary choice of resolution is not satisfactory.

To resolve this paradox, we highlight that cross-sectional structure in real-world networks typically manifests at several resolution levels simultaneously. For instance in social networks, community events of a few hours coexist with gradually evolving friendship structures (weeks to months). Similarly, in cybersecurity, malicious activity might include both rapid bursts of suspicious connections and slowly evolving patterns designed to evade detection [35]. On the other hand, in bike-sharing networks [11], interaction patterns exhibit daily rhythms (commuting), weekly cycles (workday vs. weekend usage), and seasonal trends (weather effects).

In this paper, we introduce ANIE (**A**daptive **N**etwork **I**ntensity **E**stimation), a novel approach for detecting changes in dynamic networks across multiple temporal resolutions. Our approach takes inspiration in recent work in multiresolution analysis of point process [7, 45, 14, 12], and more generally wavelet analysis [28], and adapts them to the network domain.

**Contributions** In Section 3, we formulate change detection as a statistical signal processing problem, where the goal is to recover edge-level temporal signals from noisy dynamic network observations. In Section 4, we present a new statistical method for multi-resolution change detection, supported by theoretical guarantees. In Section 5, we evaluate our method on both synthetic and real-world datasets, demonstrating that ANIE outperforms fixed-resolution approaches by effectively capturing changes at multiple time scales in dynamic networks.

## 2 Related Work

The proposed work lies at the intersection of several fields, which we briefly overview below.

**Change Detection in Dynamic Networks** The task of understanding the temporal evolution of dynamic network structure has been approached from various angles. One common approach is to view it as a dimensionality reduction task where the goal is to construct time-varying statistical summaries using for instance node embeddings [41, 29], dynamic extensions of spectral clustering [48, 31, 44, 29], latent space models [41, 42], and tensor factorization methods [25, 39, 49, 2, 17, 46], which represent temporal structure through time-evolving latent factors. A related line of work focuses specifically on detecting change points, often in an online setting, by comparing network summaries across time windows [20, 13]. These methods typically rely on fixed time intervals, which assumes short-term stationarity. We note that [50] operates in an online setting using an adapted CUSUM statistic, making direct comparison with our proposed offline method difficult.

**Wavelets and Point Process Intensity Estimation** Wavelet analysis has been proposed as a principled approach to addressing the time–frequency tradeoff in signal processing [28, 47], and has proven effective in estimating the intensity of single point processes [9, 15, 14, 12, 45, 51, 23], where key features of the intensity function often appear at multiple resolution levels. To our knowledge, our work is the first to integrate these wavelet-based point process analysis with a low-rank decomposition of cross-sectional network structure.

**Functional Data Analysis** Functional Data Analysis (FDA) has been widely used to analyze data with a continuous time dimension [40], and has been extended to multivariate settings [18]. Recent work has also adapted FDA to point process observations [37]. This work is the first to explicitly apply similar techniques to analyzing the temporal structure of dynamic networks.

# 3 Multiresolution Change Detection in Dynamic Networks

This section gives some context to our proposed method, by casting the problem of detecting significant change in dynamic networks as a network intensity estimation problem.

## 3.1 A Low-Rank Poisson Process Model

The work considers **dynamic network data**, represented by an ordered sequence of interaction events $\mathcal{E} = \{(u_m, v_m, t_m)\}_{m=1}^{M}$, where the $m$-th event represents an interaction between nodes $u_m, v_m$ belonging to a set of nodes $\mathcal{U} \triangleq \{1, \ldots, N\}$ at time $t_m$, and the timestamps are provided in increasing order $0 < t_1 < \ldots < t_M < T$. We represent this data more concisely using a matrix of counting measures $\mathbb{Y} = (\mathbb{Y}_{uv})_{u,v \in \mathcal{U}^2}$, named the **adjacency measure**, and defined on the Borel $\sigma$-algebra $\mathcal{B}(\mathcal{T})$ of the time interval $\mathcal{T} = [0, T]$. For any Borel set $\mathcal{I} \subset \mathcal{T}$, the element $\mathbb{Y}_{uv}(\mathcal{I}) = \int_{\mathcal{I}} d\mathbb{Y}_{uv}(t) = \sum_{t \in \mathcal{E}_{uv}} \mathbb{1}_{\mathcal{I}}(t) \in \mathbb{N}$ of the matrix $\mathbb{Y}(\mathcal{I})$ counts the number of interactions between nodes $u$ and $v$ that occurred within time interval $\mathcal{I}$. We model the edge-level interactions as arising from independent Inhomogeneous Poisson Processes. Mathematically, this means that there exists a matrix of absolutely continuous **intensity measures** $\mathbb{A} = (\mathcal{I} \mapsto \mathbb{A}_{uv}(\mathcal{I}))_{u,v}$ such that for any Borel set $\mathcal{I} \subset \mathcal{T}$, the count of interactions between any node pair $u, v$ on $\mathcal{I}$ is distributed as $\mathbb{Y}_{uv}(\mathcal{I}) \sim \text{Poisson}(\mathbb{A}_{uv}(\mathcal{I}))$. We denote this using the shorthand notation $\mathbb{Y} \sim \text{PoissonProcess}(\mathbb{A})$. This work relies critically on a low-rank assumption, where we assume that the interactions between nodes may be explained by means of a measure of affinity between unobserved latent factors over time. We formalize this intuition in the following definition.

**Definition 3.1** (Common Subspace Independent Processes (COSIP))**.** *A dynamic network $\mathbb{Y}$ is said to follow the **COSIP** model, i.e. $\mathbb{Y} \sim COSIP(\mathbf{U}, \mathbb{S})$ if $\mathbb{Y} \sim PoissonProcess(\mathbb{A})$, and for all borelian $\mathcal{I} \subset \mathcal{T}$, $\mathbb{A}(\mathcal{I}) = \mathbf{U}\mathbb{S}(\mathcal{I})\mathbf{U}^{\top}$ where $\mathbf{U} \in \mathbb{R}^{N \times D}$ is a **subspace matrix** whose $D$ columns are orthonormal, $\mathbb{S}(\mathcal{I}) \in \mathbb{R}^{D \times D}$ is a low-dimensional matrix measure, named the **affinity measure**, and $D$ is a latent dimension, or rank of the model.*

This model extends the COSIE model from [4] to the continuous time setting. A special case of COSIP is the **Dynamic Stochastic Block Model** (DSBM), where $\mathbf{U} \in \{0, 1\}^{N \times D}$ is a community assignment matrix, and $\mathbb{S}(\mathcal{I})$ specifies interaction rates between blocks. The COSIP model doesn't restrict the subspace matrix $\mathbf{U}$ to be binary, but assumes that the dynamic network distribution globally has low-rank. While the model is defined in terms of the measures $\mathbb{A}$ and $\mathbb{S}$, both of them are assumed to admit respective densities $\mathbf{\Lambda}(t)$ and $\mathbf{S}(t)$ with respect to Lebesgue measure on $\mathcal{T}$, such that for any Borel set $\mathcal{I} \subset \mathcal{T}$, $\mathbb{A}(\mathcal{I}) = \int_{\mathcal{I}} \mathbf{\Lambda}(t)dt$ and $\mathbb{S}(\mathcal{I}) = \int_{\mathcal{I}} \mathbf{S}(t)dt$. We refer to them as the **intensity function** and the **affinity function** respectively.

## 3.2 Change Detection as an Intensity Estimation Problem

A naive approach to estimating the intensity function $\mathbf{\Lambda}_{uv}(t)$ is to use a histogram-based estimator such as $\hat{\mathbf{\Lambda}}_{uv}(t) = B \sum_{b=1}^{B} \mathbf{1}_{\mathcal{I}_b}(t) \mathbb{Y}_{uv}(\mathcal{I}_b)$ where $\{\mathcal{I}_b\}_{b=1}^{B}$ is a partition of the time interval $[0, T]$. However, such a naive edge-level estimator will tend to reflect not only meaningful structural changes, but also spurious fluctuations due to sparsity and edge-level randomness. In contrast, under the COSIP model, the observed data $\mathbb{Y}$ is viewed as a *noisy observation* of a latent intensity measure $\mathbb{A}$, whose density $\mathbf{\Lambda}(t)$ is decomposed into a node-level subspace matrix $\mathbf{U}$ and a time-varying affinity function $\mathbf{S}(t)$, and the intensity function is expressed as a sum over pairs of latent factors, thus borrowing strength across all node pairs:

$$\mathbf{\Lambda}_{uv}(t) = \sum_{p,q \in [D]^2} \mathbf{U}_{up}\, \mathbf{U}_{vq}\, \mathbf{S}_{pq}(t).$$

Crucially, this formulation unifies the two central goals of our work. First, estimating $\mathbf{U}$ reveals the network's *cross-sectional structure*—a set of latent factors that capture how nodes align in their behavior over time. Second, estimating the time-varying affinity $\mathbf{S}(t)$ entails the identification of *structural change points* corresponding to features—for instance abrupt shifts or singularities—in the temporal signal $\mathbf{S}(t)$. In this way, detecting changes in network structure is naturally framed as the problem of *identifying meaningful temporal features of the affinity function*. As discussed in the introduction, such features often manifest at multiple resolution levels, motivating the use of wavelets for their detection.

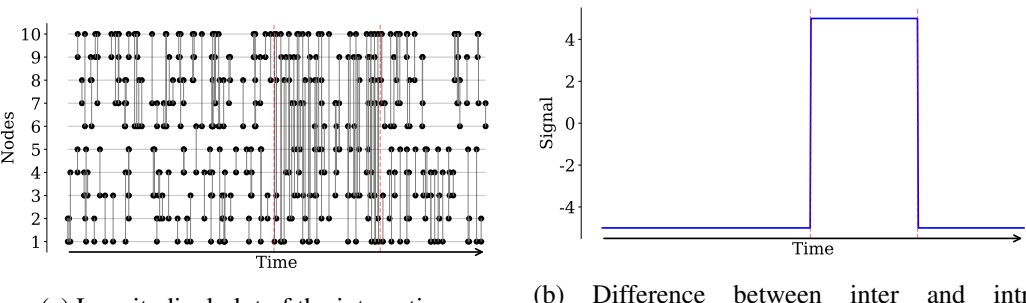

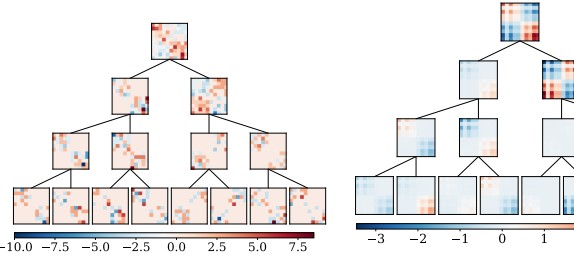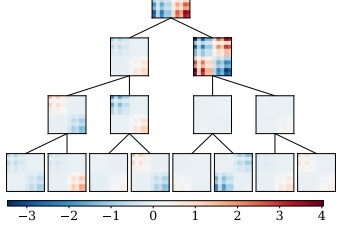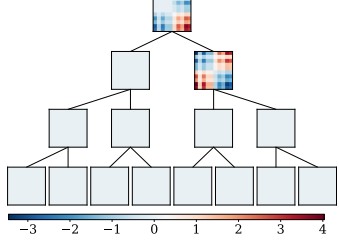

(a) Longitudinal plot of the interactions.

(b) Difference between inter and intra-community rate over time.

(c) Step 1: Hierarchy of network Haar wavelet coefficients.

(d) Step 2: Low-rank approximation of the wavelet coefficients.

(e) Step 3: Coefficients after statistical thresholding.

Figure 1: Visualization of the ANIE approach with the Haar wavelet on a dynamic stochastic block model: (a) shows the raw interaction data over time; (b) illustrates the intensity gap between intra-community and inter-community node-pairs; (c) shows the wavelet decomposition of the dynamic network, with each row representing a time scale and each leaf corresponding to a specific time location; (d) shows the low-rank approximation of the wavelet coefficients; finally (e) illustrates the denoising step where statistical thresholding is applied to the coefficients, separating the noise (bottom coefficients in (d)) from the signal (the top right coefficients in (d)).

## 4 ANIE : Adaptive Network Intensity Estimation

We now introduce ANIE (Adaptive Network Intensity Estimation), a novel method estimating the intensity measure of dynamic networks under the COSIP model by detecting significant changes in the affinity function. The method takes as input a dynamic network represented by its corresponding adjacency measure $\mathbb{Y}$, and outputs a subspace matrix $\mathbf{U}$ and an adaptive intensity estimate $\hat{\mathbf{\Lambda}}(t)$. A full algorithmic description of the procedure is provided in the appendix.

### 4.1 Function Spaces and Basis Decomposition

ANIE leverages an orthonormal functional basis $\{\phi^b\}_{b=1}^B$ of the set of square-integrable functions $L^2(\mathcal{T})$. For any measure $\mu$ on $\mathcal{T}$ and function $f$, we denote by $\mu(f) = \int_{\mathcal{T}} f(t)d\mu(t)$ the projection of $\mu$ onto $f$. When $\mu$ admits a density $\lambda(t)$ that can be decomposed as $\lambda(t) = \sum_{b=1}^B \beta^b \phi^b(t)$ in this basis, orthonormality implies that the coefficients can be obtained using projection $\beta^b = \mu(\phi^b)$. In particular for a Poisson Process $\mathbb{Y}$ with intensity measure $\mu$, the coefficients $\mathbb{Y}(\phi^b)$ provide unbiased estimates of $\beta^b$, specifically $\mathbb{E}[\mathbb{Y}(\phi^b)] = \beta^b$. This is also valid for the matrix Poisson Process $\mathbb{Y}$ considered in this paper. As such, we denote $\mathbb{A}(\phi^b)$ for the coefficients of the intensity on the basis and $\mathbb{Y}(\phi^b)$ for their empirical estimates.

**Choice of Basis** While any basis of $L^2(\mathcal{T})$ can be used, we illustrate our method using wavelet basis functions, which are known for their effectiveness in adaptive denoising [45, 28, 15]. For a non-orthonormal basis $\{\phi^b\}_{b=1}^B$ spanning $L^2(\mathcal{T})$, we can orthonormalize it using the Gram matrix $\mathbf{G} = \left(\int_{\mathcal{T}} \phi^k(t)\phi^l(t)dt\right)_{k,l=1}^B$. Indeed, defining $\Phi(t) \stackrel{\Delta}{=} [\phi^1(t), \dots, \phi^B(t)]^\top$, the rows of the vector

$\tilde{\Phi}(t) \triangleq \mathbf{G}^{-1/2}\Phi(t)$ form an orthonormal basis that can be used directly in our framework. As a result our proposed method is highly flexible and variants of it can be derived using any functional bases in $\mathcal{L}^2(\mathcal{T})$ used in functional data analysis [40]. For example, **orthonormal bases** include the Fourier basis, wavelet bases (Haar, Daubechies) and Legendre polynomials. On the other hand, **non-orthonormal bases** include B-splines, natural and cubic splines, classical polynomial bases (which can be orthonormalized as needed using the previous remark).

**Haar Wavelet Basis** We use the *Haar wavelet basis* to illustrate the multi-resolution capabilities of ANIE . This basis consists of the family $\{f\} \cup \{\psi_{j,k} \mid j \geq 0, k = 0, \ldots, 2^j - 1\}$, where the scaling function is $f(t) = \mathbb{1}_{[0,1]}(t)$ and each $\psi_{j,k}$ is a scaled and translated version of the mother wavelet $\psi(t) = \mathbb{1}_{[0,1/2)}(t) - \mathbb{1}_{[1/2,1]}(t)$, given by $\psi_{j,k}(t) = 2^{j/2}\psi(2^j t - k)$. Here, $j$ controls the scale (resolution) and $k \in \{0, \ldots, 2^j - 1\}$ the location for a given scale $j$. As shown in [45], for a dyadic interval $\mathcal{I}_{j,k} = [2^{-j}k, 2^{-j}(k+1)]$ of width $2^{-j}$, the coefficients $\mathbb{Y}(\psi_{j,k})$ measure the scaled difference in the number of events between its left and right halves:

$$\mathbb{Y}(\psi_{j,k}) = 2^{-j/2}\big[\mathbb{Y}(\mathcal{I}_{j+1,2k}) - \mathbb{Y}(\mathcal{I}_{j+1,2k+1})\big].$$

These coefficients capture changes in the empirical event intensity across scales and locations, resulting in a hierarchy of coefficient matrices as shown on Figure 1. Positive values in these matrices indicate fewer events in the right subinterval, while negative values indicate more. As commonly done in wavelet analysis, we use a finite subset of this basis up to a maximum scale $J$.

**Note:** Throughout the paper, we use $\phi^{(b)}$ (indexed by $b$) to denote a generic basis function. For the Haar basis, this set includes the scaling function $f$ and all wavelet functions $\psi_{j,k}$ with $j \geq 0$ and $k \in \{0, \ldots, 2^j - 1\}$, with $b$ serving as a single unified index over them.

## 4.2   First stage: Low-Rank Decomposition

**Basis Decomposition** The first step of ANIE decomposes the adjacency measure on the basis $\{\phi^b\}_{b=1}^B$, resulting in **empirical coefficients**: $\mathbb{Y}(\phi^b) \triangleq \int_\mathcal{T} \phi^b(t)d\mathbb{Y}(t) = \left(\sum_{\tau \in \mathcal{E}_{u,v}} \phi^b(\tau)\right)_{u,v} \in \mathbb{R}^{N \times N}$, where $\mathcal{E}_{u,v}$ is the set of interaction times between nodes $u$ and $v$. This computation is efficient: for each node pair, we evaluate the basis function at each interaction time and sum the results. Notably, the resulting coefficient matrices inherit the sparsity of the adjacency measure. Moreover, by Campbell's theorem [5], these coefficients are unbiased estimates of the coefficients of the true intensity: $\mathbb{E}[\mathbb{Y}(\phi^b)] = \mathbb{A}(\phi^b)$.

**Global Subspace Estimation** The estimation of the subspace U follows closely the UASE [21] strategy. The empirical coefficients are first arranged into a unfolded matrix

$$\mathbf{X} = [\mathbb{Y}(\phi^1)^T \| \mathbb{Y}(\phi^2)^T \| \cdots \| \mathbb{Y}(\phi^B)^T] \in \mathbb{R}^{N \times NB}$$

whose rows represent each node's relational behaviors over time. Despite their high dimensionality, these behaviors typically exhibit low-dimensional structure due to two alignment factors: *cross-sectional alignment* (often reflecting community structure) and *longitudinal alignment* (reflecting structure in nodes' activity patterns). For example, in a social network, nodes may interact with the same communities but at different times, placing them in related but distinct behavioral subspaces. To capture these dominant modes of variation, we apply Truncated Singular Value Decomposition (TSVD) to matrix $\mathbf{X}$, extracting the $D$ singular vectors corresponding to the largest singular values into a matrix $\hat{\mathbf{U}} \in \mathbb{R}^{N \times D}$. As a note, this step may be viewed as a partial Tucker decomposition of the 3-mode tensor $\{\mathbb{Y}_{uv}(\phi^b)\}$, where the first two modes are the node indices and the third mode is the basis index, namely a SVD of the mode-2 unfolding of the tensor [39]. Under suitable assumptions, this subspace estimation procedure is consistent, as formalized in the following theorem.

**Theorem 4.1** (Subspace Estimation Consistency ). *Suppose that $\mathbb{Y} \sim COSIP(\mathbf{U}, \mathbb{S})$ and that there exists a fixed matrix-function $\mathbf{R}(t) = \sum_{b=1}^B \mathbf{C}^b \phi^b(t) \in \mathbb{R}^{D \times D}$, and a sparsity factor $\rho_N \leq 1$ satisfying $N\rho_N = \omega(\log^3(N))$, such that $\mathbf{S}(t) := \rho_N \mathbf{R}(t)$. In addition, suppose that*

1. *The matrix $\mathbf{\Delta} = \sum_{b=1}^B (\mathbf{C}^b)^\top \mathbf{C}^b$ has full rank.*

2. *The subspace matrix $\mathbf{U}$ satisfies the incoherence condition*

$$\|\mathbf{U}\|_{2,\infty} = O\left(\sqrt{\tfrac{\log(N)}{N\rho_N}}\right).$$

*Then, there exists an orthogonal matrix* $\mathbf{Q}$ *such that*

$$\|\hat{\mathbf{U}}\mathbf{Q} - \mathbf{U}\|_2 = \mathcal{O}_{\mathbb{P}}\left(\frac{1}{\sqrt{N\rho_N}}\right) \tag{1}$$

A proof of Theorem 4.1 is provided in the appendix.

## 4.3 Second stage: Denoising through statistical thresholding

The first stage of ANIE outputs an estimated subspace matrix $\hat{\mathbf{U}} \in \mathbb{R}^{N \times D}$, which encodes the cross-sectional structure by representing each node in terms of its projection onto the dominant latent factors of node behavior. Based on this, it is natural to consider that each pair $p, q$ of these latent factors will be subject to change over time. This change can be quantified directly by combining the coefficients of all the node pairs and weighing them by their respective nodes' affinity with the latent factors, which we do here:

**Definition 4.1** (Empirical affinity coefficients)**.** *The **empirical affinity coefficients** are defined as the collection of* $D \times D$ *matrices*

$$\hat{\mathbb{S}}(\phi^b) = \hat{\mathbf{U}}^T \mathbb{Y}(\phi^b)\hat{\mathbf{U}} \in \mathbb{R}^{D \times D}, \quad \forall b \in [B]. \tag{2}$$

The empirical affinity coefficients play a central role in our approach and offer several advantages. First, they have well-defined statistical properties: their expectation is $\mathbb{E}[\hat{\mathbb{S}}_{pq}(\phi^b)] = \sum_{u,v} \hat{\mathbf{U}}_{up}\hat{\mathbf{U}}_{vq}\mathbf{\Lambda}_{uv}(\phi^b)$, and their variance is $\mathrm{Var}[\hat{\mathbb{S}}_{pq}(\phi^b)] = \sum_{u,v} \hat{\mathbf{U}}_{up}^2 \hat{\mathbf{U}}_{vq}^2 \mathbf{\Lambda}_{uv}((\phi^b)^2)$. These results follow directly from the distributional properties of point process projections [23]. Moreover, when using Haar wavelet functions $\phi^b = \psi_{jk}$, they have a straightforward interpretation: each $\hat{\mathbb{S}}_{pq}(\psi_{jk})$ captures changes in interaction intensity between latent factors $p$ and $q$ at specific scales and locations. Large values indicate potential structural changes over the support of $\psi_{jk}$. For instance, with the Haar wavelet, positive (respectively negative) coefficients correspond to decreasing (respectively increasing) interaction affinity between factors $p$ and $q$ over the interval $\mathcal{I}_{j,k}$. Extreme values reflect strong structural changes between latent factors. Finally, due to the fact that they borrow strength across node pairs, the next theorem shows that they are asymptotically normal under suitable conditions, enabling statistical testing.

**Theorem 4.2** (Asymptotic normality of the empirical affinity coefficients)**.** *Suppose that there exists sequences* $\alpha_N, \beta_N, \mu_N$ *such that for all* $u, v \in [N]$, $p \in [D]$ *and* $t \in \mathcal{T}$,

$$0 < \alpha_N \leq \mathbf{\Lambda}_{uv}(t) \leq \beta_N \qquad and \qquad \hat{\mathbf{U}}_{up}^2 \leq \frac{\mu_N}{N}$$

*which satisfy*

$$\frac{\mu_N^3}{N}\left(\frac{\beta_N}{\alpha_N}\right)^{3/2} \to 0 \qquad as\ N \to \infty.$$

*Then, the standardized version of the empirical affinity coefficients* $\hat{\mathbb{S}}_{pq}(\phi^b)$ *defined in 4.1 converge to a standard normal distribution as* $N \to \infty$. *More specifically:*

$$\frac{\hat{\mathbb{S}}_{pq}(\phi^b) - \mathbb{E}[\hat{\mathbb{S}}_{pq}(\phi^b)]}{\sqrt{\mathrm{Var}[\hat{\mathbb{S}}_{pq}(\phi^b)]}} \xrightarrow{d} \mathcal{N}(0,1), \qquad as\ N \to \infty. \tag{3}$$

The proof of Theorem uses the Lyapunov Central Limit Theorem applied to the family of independent variables $\{\hat{\mathbf{U}}_{up}\hat{\mathbf{U}}_{vq}\mathbb{Y}(\phi^b)\}_{u,v}$ for a given $b$, and is included in the appendix.

**Multiple statistical testing for change in the network structure**

As a result of Theorem 4.3, the task of identifying changes in the network structure can be formulated as a multiple statistical testing problem, where the null hypotheses are that the latent factors $p$ and $q$ are not significantly associated with the wavelet functions $\phi^b$: $\mathcal{H}_{p,q}^b = \mathbb{E}[\hat{\mathbb{S}}_{pq}(\phi^b)] = 0$. To carry out these tests, we may define the following Z-scores, by replacing the expectation by 0, and the variance by its empirical estimate:

$$\mathbf{Z}_{pq}(\phi^b) = \frac{\hat{\mathbb{S}}_{pq}(\phi^b)}{\sqrt{\tilde{\mathrm{Var}}[\hat{\mathbb{S}}_{pq}(\phi^b)]}}, \quad where \quad \tilde{\mathrm{Var}}[\hat{\mathbb{S}}_{pq}(\phi^b)] = \sum_{u,v} \hat{\mathbf{U}}_{up}^2 \hat{\mathbf{U}}_{vq}^2 \hat{\mathbb{Y}}_{uv}\left((\phi^b)^2\right). \tag{4}$$

Under the null hypothesis that $\mathbb{E}[\hat{\mathbb{S}}_{pq}(\phi^b)] = 0$ (indicating no correlation with $\phi^b$), these Z-scores follow approximately a standard normal distribution $\mathcal{N}(0, 1)$ for large $N$. This is particularly relevant when using wavelets $\phi^b = \psi_{jk}$, as the null hypothesis corresponds to a locally constant intensity between a pair of factors, $p$ and $q$, on the support of the wavelet. To determine which coefficients are statistically significant, we compare $|\mathbf{Z}_{pq}(\phi^b)|$ with a threshold. Since we conduct $B \times D \times D$ simultaneous tests (one for each coefficient), we must account for multiple comparisons. We control the False Discovery Rate (FDR) using the Benjamini-Hochberg procedure [8] at a significance level $\alpha$ (typically 0.05), resulting in a binary significance mask $M_{pq}^b \in \{0, 1\}$. The final denoised intensity estimate is then constructed using only the coefficients determined to be statistically significant. We note that there exist more approaches for thresholding wavelet coefficients [24, 15, 45] which we didn't explore in this work but could improve the accuracy of the thresholding stage.

### 4.4 Parameter Selection and Computational Efficiency

The rank $D$ can be determined by examining the scree plot of singular values of the matrix $\mathbf{X}$. In turn, the choice of significance level $\alpha$ reflects how concervative/agressive the thresholding should be. A threshold $\alpha = 0.0$ will lead to a constant signal, since all the detail coefficients will be classified as noise. Conversely, a threshold $\alpha = 1.0$ will classify all the coefficients as signal, leading to a noisy estimate. Typically, $\alpha$ is set to $0.05$, as is common in multiple testing scenarios. However, the choice of $\alpha$ can be adjusted based on the specific application and desired level of significance. Our method can be made time and memory efficient by leveraging sparsity in three key ways: (1) coefficients $\mathbb{Y}(\phi^b)$ naturally inherit the sparsity of the original adjacency measure $\mathbb{Y}$, (2) SciPy's sparse SVD implementation can be employed compute dominant singular vectors without constructing dense matrices, and (3) thresholding operates only on the compact $D \times D$ affinity matrices $\hat{\mathbb{S}}(\phi^b)$ rather than the full $N \times N$ matrices $\mathbb{Y}(\phi^b)$. A timed experiment reporting computation times for varying numbers of nodes is provided in the appendix. All our experiments were run on a MacBook Air with a M1 chip and 8GB of RAM.

## 5 Experiments

The proposed ANIE method is evaluated on two tasks. First, we generate synthetic Erdős-Renyi (ER) and Stochastic Block Model (SBM) datasets, and measure the performance of our method in estimating a known network intensity from an observed dynamic network. Second, in order to demonstrate the practical utility of our method, we apply it to the task of detecting change points in a real-world dataset of message interactions, and compare our method with two existing methods: Laplacian Anomaly Detection (LAD) [20] and Tensorsplat [26].

### 5.1 Intensity Estimation on Synthetic Datasets

**Dataset** We test our wavelet-based approach on synthetic datasets designed specifically to test temporal adaptivity. We simulate networks with both Erdős-Rényi (ER) and Stochastic Block Model (SBM) structures, where intensity functions show complex temporal patterns. For ER-blocks, every node pair shares the same intensity, $\mathbf{\Lambda}_{uv}(t) = \lambda_{blocks}(t)$, based on the "blocks" test function from [15], which features blocks of varying widths. For SBM, we generate a two-community network with piecewise constant intensities: intra-community intensities are significantly higher than inter-community ones, except over an interval where they are equal. This setting tests our method's ability to detect sharp intensity changes. For the Erdős-Renyi model, we generate various networks with a number of nodes ranging from 50 to 1000. In contrast for the SBM model, we generate networks with 50 to 2500 nodes.

**Experimental Setting** We compare our ANIE-Haar approach against non-adaptive IPP estimators from [29]. IPP first constructs a naive intensity estimate $\tilde{\mathbf{\Lambda}}$, then applies low-rank denoising $\hat{\mathbf{\Lambda}}(t) = \hat{\mathbf{U}}\hat{\mathbf{U}}^T\tilde{\mathbf{\Lambda}}(t)\hat{\mathbf{U}}\hat{\mathbf{U}}^T$. We consider two variants: **IPP-KDE**, which constructs $\tilde{\mathbf{\Lambda}}$ using kernel density estimation, and **IPP-Hist**, which uses histogram-based estimation. Note that IPP-Hist is equivalent to ANIE with the Haar wavelet without thresholding. We use the Mean Integrated Squared Error (MISE) as our primary metric: $\text{MISE} = \frac{1}{N^2}\sum_{(u,v)\in[N]^2}\int_{\mathcal{T}}\left|\mathbf{\Lambda}_{uv}(t) - \hat{\mathbf{\Lambda}}_{uv}(t)\right|^2 dt$. This metric averages the intensity estimation error over node pairs. In our setting we average the error over patches of $N = 100$ nodes (i.e. $100^2$ node pairs). We report mean and standard error over 10 runs.

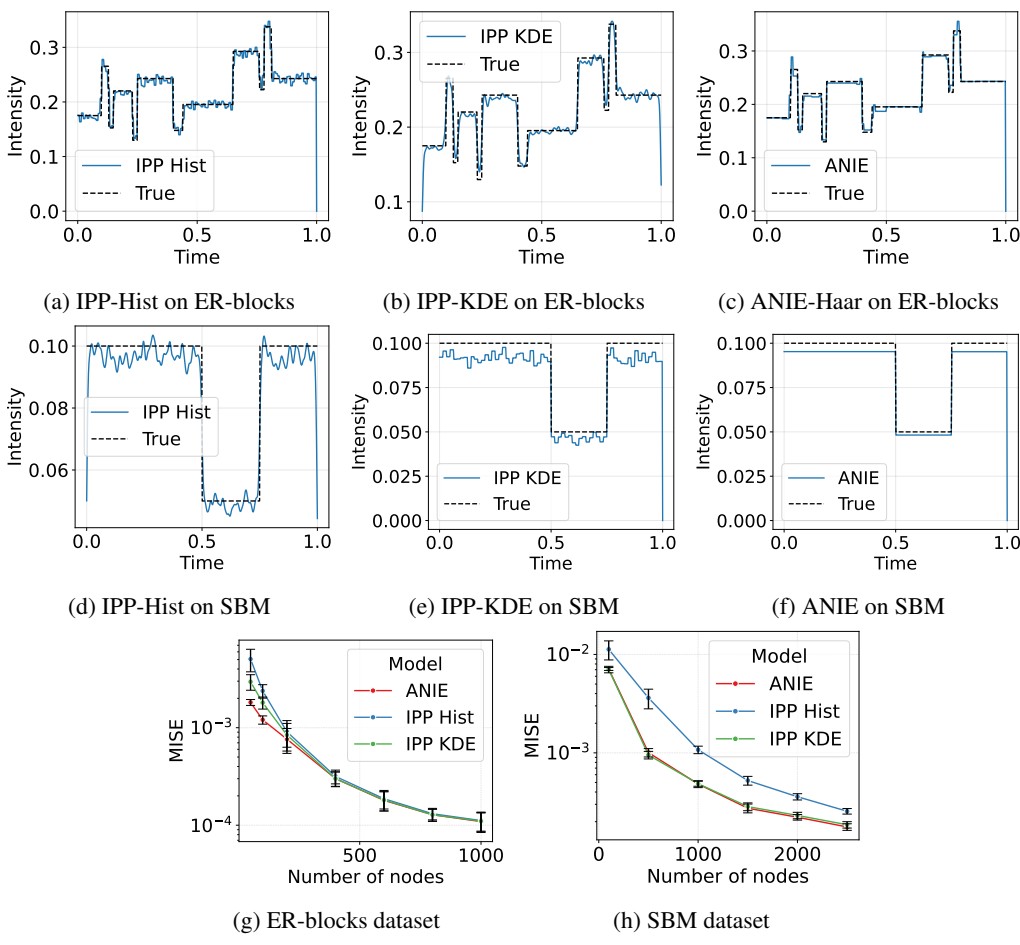

Figure 2: Comparison of intensity estimation methods on ER-blocks and SBM datasets. The first two rows show the estimated intensity functions for different methods, while the last row shows the MISE vs. number of nodes for both datasets.

**Results** The experimental results, summarized in Figure 2 demonstrate two essential advantages of ANIE . A first notable advantage is *multi-scale abilities*. As can be seen in Figure 2c that the proposed approach allows capturing perturbations of the underlying intensity which occur at different temporal resolutions, while staying robust to noise. In contrast, in order to accurately capture the same perturbations, non-adaptive methods such as IPP-Hist and IPP-RBF pay the price of overfitting to noise, leading to spurious oscillations in the flat regions of the intensity function, as can be seen in 2b, 2a. A second, related key advantage is *sharp change localization*: the ANIE approach precisely identifies structural change points in the empirical affinity coefficients (Figure 2f), while staying robust to noise. In this setting, the intensity function between node pairs is piecewise constant, with abrupt changes in the intensity function. Our method effectively captures these changes, while other methods like IPP-Hist and IPP-RBF require a small bandwidth, and thus overfit to noise, in order to capture the same changes.

## 5.2   Case Study: Multi-scale Anomaly Detection on the UCI Messages Dataset

**Dataset** This experiment evaluates the practical utility of our ANIE method for detecting change-points in real-world interaction datasets. To this end, we apply ANIE to the UCI Messages dataset, as done in previous work [20]. This dataset contains 59,835 messages sent among 1,899 users over a period of 196 days between April and October 2004. Each message interaction is represented as a directed edge with a weight corresponding to the number of characters in the message.

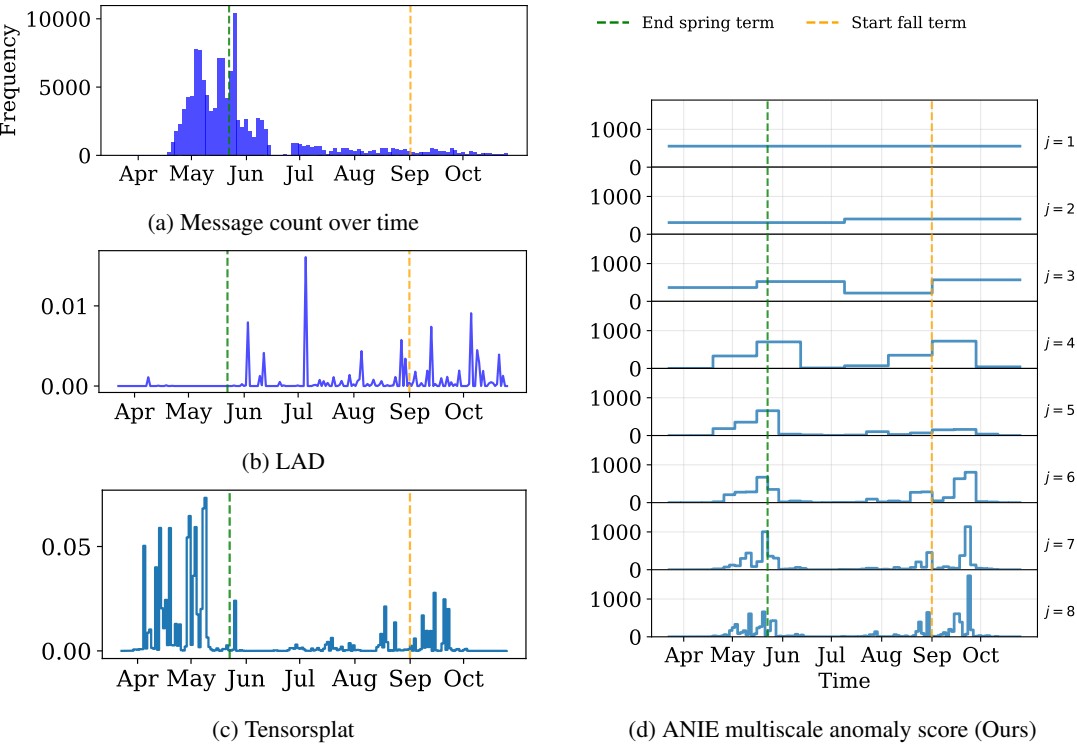

Figure 3: Comparison of anomaly detection methods on the UCI dataset. Left column show respectively (a) message count over time, (b) LAD anomaly score [20], and (c) Tensorsplat [26]. Right column (d) shows the multi-scale anomaly scores from our ANIE method. The two main events identified in [32] are highlighted with vertical dashed lines.

**Experimental Setting** We compare our ANIE method against two existing approaches for anomaly detection in temporal networks. *Laplacian Anomaly Detection* (LAD) [20] bins the data into temporal snapshots, embeds each graph using its Laplacian eigenvalues, and detects anomalies by monitoring changes in these successive spectral representations. The *Tensorsplat* [26] anomaly score is calculated as $\text{Tensorsplat}(t) = \|\mathbf{T}(t) - \mathbf{T}(t-1)\|_2$ where $\mathbf{T}(t)$ is the the row $t$ of the third matrix in the PARAFAC decomposition of the fixed-resolution adjacency tensor. In contrast, for a given scale $j$ the ANIE multi-multiscale anomaly score at scale $j$ is defined as the Frobenius norm of the $2^j$ empirical affinity : for all $k = 0, \ldots, 2^j - 1, E_j(k) \triangleq \left\| \hat{\mathbb{S}}(\psi_{j,k}) \right\|_F$. Using piecewise constant interpolation, we then convert this discrete time series into a continuous function $E_j(t)$ defined on the interval $[0, 1]$ which we plot on Figure 3d. We use the Haar Wavelet basis in this example.

**Results** Figure 3 shows anomaly detection results on the UCI Messages dataset. As can be seen, ANIE successfully identifies major structural changes in the network, notably at the end of the spring term (day 60) and the start of the fall term (day 150). We first validate that this second event is not detectable by simply counting message volume over time. Tensorsplat seems to follow to a great extent the message count over time. This is likely due to the fact that in the PARAFAC decomposition, the third factor indirectly encodes the interaction rate into the latent activity vectors. Unlike LAD, which produces a single aggregated anomaly score, ANIE provides multi-resolution scores that distinguish large-scale structural changes from finer scale temporal oscillations. As shown in Figure 3d, the two main events described in [32] lead to change observed over different time scales. A possible explanation for the discrepancy between resolutions is that coarse-scale changes (green and blue curves) capture the formation and dissolution of social groups at the end and beginning of the academic year, whereas finer-scale changes (yellow) reflect short-term fluctuations driven by the academic calendar, such as classes, group projects, or other time-limited activities.

# 6 Discussion and Conclusion

Due to the continuous nature of dynamic networks, tools from functional data analysis such as basis expansions combined with low-rank approximations appear to be a natural fit for analyzing dynamic networks. In this work, we have presented ANIE , a method that uses low-rank approximation to estimate the global structure of the data. It then employs a multi-resolution, wavelet-based approach to test for significant changes in network structure at different resolution levels. In doing so, it addresses the time–frequency trade-offs inherent to fixed-bandwidth and discretized approaches.

The proposed methodology comes with several limitations. First, the testing strategy relies on an estimate of the variance of the empirical affinity coefficients for computing the Z-scores (Equation 4). The robustness of the thresholding procedure is therefore highly sensitive to this variance estimation error, particularly at high resolutions. Exploring alternative thresholding strategies thus appears to be a promising direction for future work. Moreover, in this study, we have focused on the Haar wavelet basis due to its convenient interpretation as an adaptive histogram. However, in applications requiring smoother intensity estimates, alternative bases such as Daubechies wavelets [28] or B-spline-based approaches like Splinets [38, 27] could be employed. Additionally, our method could be extended by incorporating tensor factorization techniques such as PARAFAC [17], applied to the sparse tensor of empirical coefficients. Lastly, an interesting avenue would be to explore a hybrid Tucker–Karhunen–Loève decomposition, building on recent work in PCA for point processes [37], potentially leading to a deeper theoretical understanding of the opportunities and limitations of functional analysis on dynamic networks.

## Acknowledgments and Disclosure of Funding

The research leading to these results has received funding from the Special Research Fund (BOF) of Ghent University (BOF20/IBF/117), from the Flemish Government under the "Onderzoeksprogramma Artificiële Intelligentie (AI) Vlaanderen" programme, from the FWO (project no. G0F9816N, 3G042220, G073924N). Funded by the European Union (ERC, VIGILIA, 101142229). Views and opinions expressed are however those of the author(s) only and do not necessarily reflect those of the European Union or the European Research Council Executive Agency. Neither the European Union nor the granting authority can be held responsible for them. Modell and Heard acknowledge support from EPSRC NeST Programme grant EP/X002195/1.

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

# Appendix to the paper "Multiresolution Analysis and Statistical Thresholding on Dynamic Networks"

## A  Full algorithm

Algorithm 1 describes the full pipeline of our proposed Adaptive Network Intensity Estimation (ANIE) method.

---

**Algorithm 1** Adaptive Network Intensity Estimation

---

**Input:** Dynamic network $\mathbb{Y}$, orthonormal basis $\{\phi^b\}_{b=1}^B$ of $\mathcal{L}^2(\mathcal{T})$, rank $D$, significance level $\alpha$

1: **Basis decomposition:** Decompose the dynamic network on the functional basis

$$\mathbb{Y}(\phi^b) = \int_{\mathcal{T}} \phi^b(t) d\mathbb{Y}(t) \in \mathbb{R}^{N \times N}, \quad b = 1, \ldots, B$$

2: **Low rank estimation:** Form the concatenated matrix and compute truncated SVD

$$\mathbf{X} = [\mathbb{Y}(\phi^1)^T \| \mathbb{Y}(\phi^2)^T \| \cdots \| \mathbb{Y}(\phi^B)^T] \in \mathbb{R}^{N \times NB}$$

$$\mathbf{X} \approx \hat{\mathbf{U}} \boldsymbol{\Sigma} \hat{\mathbf{V}}^T \quad \text{(keeping } D \text{ largest singular values)}$$

Calculate the *empirical affinity coefficients*

$$\hat{\mathbb{S}}(\phi^b) = \hat{\mathbf{U}}^T \mathbb{Y}(\phi^b) \hat{\mathbf{U}} \in \mathbb{R}^{D \times D}$$

And their associated *sample variance estimates*

$$\tilde{\mathrm{Var}}[\hat{\mathbb{S}}_{pq}(\phi^b)] = \sum_{u,v} \hat{\mathbf{U}}_{up}^2 \hat{\mathbf{U}}_{vq}^2 \mathbb{Y}_{uv} \left((\phi^b)^2\right) \in \mathbb{R}^{D \times D}$$

3: **Statistical thresholding:** For each $p, q \in [D]^2$ and $b \in [B]$, compute

$$\text{Z-score } Z_{pq}^b = \frac{\hat{\mathbb{S}}_{pq}(\phi^b)}{\sqrt{\tilde{\mathrm{Var}}[\hat{\mathbb{S}}_{pq}(\phi^b)]}}, \quad \text{and associated p-value } p_{pq}^b = 2\left(1 - \Phi(|Z_{pq}^b|)\right),$$

where $\Phi$ is the standard normal cumulative density function. Then apply multiple-testing correction to obtain the corrected p-values $\tilde{p}_{pq}^b$, and finally the thresholded coefficients using

$$T^\alpha\left(\hat{\mathbb{S}}_{pq}(\phi^b)\right) = \begin{cases} \hat{\mathbb{S}}_{pq}(\phi^b), & \text{if the coefficient is significant, i.e. } \tilde{p}_{pq}^b < \alpha, \\ 0, & \text{otherwise.} \end{cases}$$

4: **Reconstruction:** Compute thresholded intensity estimate

$$\hat{\boldsymbol{\Lambda}}(t) = \hat{\mathbf{U}} \left(\sum_{b=1}^B T^\alpha(\hat{\mathbb{S}}(\phi^b))\phi^b(t)\right) \hat{\mathbf{U}}^T$$

**Output:** Low-rank subspace $\hat{\mathbf{U}}$, significance mask $\mathcal{M}_{pq}^b = \mathbb{1}_{\{\tilde{p}_{pq}^b < \alpha\}}$, intensity estimate $\hat{\boldsymbol{\Lambda}}(t)$.

---

## B  Proof of theorem 4.1

Recall that

$$\mathbf{X} = [\mathbb{Y}(\phi^1)^T \| \mathbb{Y}(\phi^2)^T \| \cdots \| \mathbb{Y}(\phi^B)^T] \in \mathbb{R}^{N \times nN}$$

and then by the properties of Poisson processes, we have that

$$\mathbb{E}\mathbf{X} = [\mathbb{A}(\phi^1)^T \| \mathbb{A}(\phi^2)^T \| \cdots \| \mathbb{A}(\phi^B)^T] \in \mathbb{R}^{N \times nN}$$

Observe that

$$\mathbb{A}(\phi^b) = \int_{\mathcal{T}} \mathbf{\Lambda}(t)\phi^b(t)dt = N\rho_N \int_{\mathcal{T}} \mathbf{U}\mathbf{R}(t)\mathbf{U}^\top\phi^b = N\rho_N \mathbf{U}\left(\int_{\mathcal{T}} \mathbf{R}(t)\phi^b(t)dt\right)\mathbf{U}^\top.$$

In addition, since the basis functions $\phi^1, \ldots, \phi^B$ are orthonormal, we have that

$$\int_{\mathcal{T}} \mathbf{R}(t)\phi^b dt = \int_{\mathcal{T}} \left(\sum_{b'=1}^{B} \mathbf{C}^{b'}\phi^{b'}\right)dt = \sum_{b'=1}^{B} \mathbf{C}^{b'}\left(\int_{\mathcal{T}} \phi^{b'}(t)\phi^b(t)dt\right) = \mathbf{C}^b.$$

It follows that $\mathbb{A}(\phi^b) = N\rho_N \mathbf{U}\mathbf{C}^b\mathbf{U}^\top$ and

$$\mathbb{E}\mathbf{X} = N\rho_N \mathbf{U}\mathbf{C}\mathbf{U}^\top, \qquad \mathbf{C} = [\mathbf{C}^1 \| \mathbf{C}^2 \| \cdots \| \mathbf{C}^B].$$

Therefore, there exists an orthogonal matrix $\mathbf{O}_1 \in \mathbb{R}^{D \times D}$ such that the left (orthonormal) singular values corresponding to the non-zero singular values of $\mathbb{E}\mathbf{X}$, which we denote $\sigma_1 \geq \cdots \geq \sigma_D$, are given by the columns of $\mathbf{U}\mathbf{O}_1$.

Let $\hat{\mathbf{U}} = (\hat{u}_1, \ldots, \hat{u}_D)$ be the matrix whose columns contains the left (orthonormal) singular vectors of $\mathbf{X}$ corresponding to the $D$ largest eigenvalues of $\mathbf{X}$, which we denote $\hat{\sigma}_1 \geq \cdots \geq \hat{\sigma}_D$.

Then, by Wedin's $\sin\Theta$ theorem [10, Theorem 2.9] we have, providing $\|\mathbf{X} - \mathbb{E}\mathbf{X}\|_2 \leq (1 - 1/\sqrt{2})(\sigma_D - \sigma_{D+1})$ that there exists an orthogonal matrix $\mathbf{O}_2 \in \mathbb{R}^{D \times D}$ such that

$$\left\|\hat{\mathbf{U}} - \mathbf{U}\mathbf{O}_1\mathbf{O}_2\right\|_2 \leq \frac{\|\mathbf{X} - \mathbb{E}\mathbf{X}\|_2}{\sigma_D - \sigma_{D+1}}. \tag{5}$$

By assumption, the matrix $\mathbf{\Delta} = \sum_{b=1}^{B} (\mathbf{C}^b)^\top \mathbf{C}^b$ has full rank, and therefore $\sigma_1, \ldots, \sigma_D = \Theta(N\rho_N)$. The matrix $\mathbb{E}\mathbf{X}$ has rank $D$ and so $\sigma_{D+1} = 0$. Therefore $\sigma_D - \sigma_{D+1} = \Omega(N\rho_N)$.

To complete the proof, it will suffice to show that $\|\mathbf{X} - \mathbb{E}\mathbf{X}\|_2 = \mathcal{O}_\mathbb{P}(\sqrt{N\rho_N})$, after which we can subsequently right-multiply equation 5 by $\mathbf{Q} := (\mathbf{O}_1\mathbf{O}_2)^\top$ to conclude the proof.

To do so, we will prove the following concentration inequality, which we prove in Section B.1.

**Lemma 1.** *Let $\mathbb{Y}$ denote the counting measure of an inhomogeneous Poisson process with finite intensity measure $\mathbb{A}$ on $[0,1)$. Let $\phi \in \mathcal{L}^2([0,1))$ and let $L$ be a value such that $\phi(t) \leq L$. Then*

$$\mathbb{P}\left(|\mathbb{Y}(\phi) - \mathbb{A}(\phi)| > t\right) \leq 2\exp\left\{-\frac{t^2}{2(\mathbb{A}(\phi) + tL/3)}\right\}.$$

*In particular, for $t \geq \mathbb{A}(\phi)$,*

$$\mathbb{P}\left(|\mathbb{Y}(\phi) - \mathbb{A}(\phi)| > t\right) \leq 2\exp\left(-\frac{3t}{8L}\right)$$

In particular, since $\|\mathbf{U}\|_{2,\infty} = \mathcal{O}(\sqrt{\log(N)/N\rho_N})$ by assumption, we have that $\mathbb{A}_{uv}(\phi^b) = \mathcal{O}(\log(N))$ and therefore by Lemma 1 we have that $|\mathbb{Y}_{uv}(\phi^b) - \mathbb{A}_{uv}(\phi^b)| = O_\mathbb{P}(\log(N))$. By a union bound, this holds *simultaneously* for all $u, v \in \{1, \ldots, N\}, b \in \{1, \ldots, B\}$.

To obtain a bound on $\|\mathbf{X} - \mathbb{E}\mathbf{X}\|_2$, we observe that this is equal to $\|\mathbf{E}\|_2$ where $\mathbf{E}$ is the *symmetric dilation* of $\mathbf{X} - \mathbb{E}\mathbf{X}$ [19, Theorem 7.3.3]. I.e.

$$\mathbf{E} = \begin{pmatrix} \mathbf{0} & \mathbf{X} - \mathbb{E}\mathbf{X} \\ (\mathbf{X} - \mathbb{E}\mathbf{X})^\top & \mathbf{0} \end{pmatrix}.$$

We then apply the following concentration inequality for random symmetric matrices to $\mathbf{E}$ which is Corollary 3.12 of Bandeira and Van Handel [6].

**Lemma 2** (Corollary 3.12 of Bandeira and Van Handel [6]). *Let $\mathbf{M}$ be an $N \times N$ symmetric matrix whose entries $m_{ij}$ are independent random variables which obey*

$$\mathbb{E}(m_{ij}) = 0, \qquad |m_{ij}| \leq L, \qquad \sum_{j=1}^{N} \mathbb{E}(m_{ij}^2) \leq \nu$$

*for all $i, j$. There exists a universal constant $\tilde{C} > 0$ such that for any $t \geq 0$,*

$$\mathbb{P}\left\{\|\mathbf{M}\|_2 \geq 3\sqrt{\nu} + t\right\} \leq N \exp\left(-\frac{t^2}{\tilde{C}L^2}\right).$$

We then apply Lemma 2 to $\mathbf{E}$, conditional on $\mathbf{E}_{uv} = \mathcal{O}(\log(N))$ (which holds with overwhelming probability due to the above derivation) with $L = \mathcal{O}(\log(N))$ and $\nu = \mathcal{O}(BN\rho_N) = \mathcal{O}(N\rho_N)$ (since $B$ is assumed to be fixed). We obtain

$$\|\mathbf{X} - \mathbb{E}\mathbf{X}\|_2 = \|\mathbf{E}\|_2 = O_\mathbb{P}(\sqrt{N\rho_N} + \log^{3/2}(n)) = O_\mathbb{P}(\sqrt{N\rho_N})$$

where the final inequality follows from the assumption that $N\rho_N = \Omega(\log^3(n))$. This completes the proof.

### B.1 Proof of Lemma 1

For a given $N \in \mathbb{N}$, let $X_1^{(N)}, \ldots, X_N^{(N)}$ denote $N$ independent Poisson random variables with rates

$$\lambda_n^{(N)} := \mathbb{A}\left(\left[\frac{n-1}{N}, \frac{n}{N}\right]\right) \qquad n = 1, \ldots, N.$$

Then, by the definition of an inhomogeneous Poisson process we have that

$$\mathbb{Y}(\phi) = \lim_{N \to \infty} \sum_{n=1}^N X_n^{(N)} \phi\left(\frac{n}{N}\right).$$

In addition, by a property of Poisson random variables, we have that

$$X_n^{(N)} = \lim_{M \to \infty} \sum_{m=1}^M Y_m^{(M,N)}$$

where $Y_1^{(M,N)}, \ldots, Y_M^{(M,N)}$ are independent and identically-distributed Bernoulli random variables with success probabilities $\lambda_n^{(N)}/M$. Therefore

$$\mathbb{Y}(\phi) = \lim_{M \to \infty} \lim_{N \to \infty} \sum_{m=1}^M \sum_{n=1}^N Z_{m,n}^{(M,N)}, \qquad Z_{m,n}^{(M,N)} = Y_m^{(M,N)} \phi\left(\frac{n}{N}\right).$$

Observe that

$$\mathbb{E}Z_{m,n}^{(M,N)} = \frac{\lambda_n^{(N)}}{M} \phi\left(\frac{n}{N}\right).$$

and define $E_{m,n}^{(M,N)} = Z_{m,n}^{(M,N)} - \mathbb{E}Z_{m,n}^{(M,N)}$ which are independent zero-mean random variables. Then, we have that $\left|E_{m,n}^{(M,N)}\right| \leq L$ and for sufficiently large $M$

$$\sigma^{(M,N)} := \sum_{m=1}^M \sum_{n=1}^N \mathbb{E}\left\{\left(E_{m,n}^{(M,N)}\right)^2\right\} \leq \sum_{m=1}^M \sum_{n=1}^N Z_{m,n}^{(M,N)}.$$

Note that taking limits on both sides we have that $\lim_{M \to \infty} \lim_{N \to \infty} \sigma^{(M,N)} = \mathbb{A}(\phi)$.

Now, by Bernstein's inequality, we have that

$$\sum_{m=1}^M \sum_{n=1}^N \mathbb{P}\left(\left|E_{m,n}^{(M,N)}\right| > t\right) \leq 2 \exp\left\{-\frac{t^2}{2(\sigma^{(M,N)} + t\phi_{\max}/3)}\right\}.$$

Taking $M \to \infty$ and $N \to \infty$ on both sides, we obtain the desired bound.

## C   Proof of Theorem 4.2

**Lemma 3** (Lyapunov's Central Limit Theorem (CLT) ). *Let $X_1, \ldots, X_n$ be independent random variables with finite mean and variance. If for some $\delta > 0$:*

$$\frac{1}{s_n^{2+\delta}} \sum_{i=1}^{n} \mathbb{E}\left[|X_i - \mathbb{E}[X_i]|^{2+\delta}\right] \xrightarrow[n \to \infty]{} 0$$

*where $s_n^2 = \sum_{i=1}^{n} \mathrm{Var}[X_i]$ is the cumulated variance, then the sum $S_n = \sum_{i=1}^{n} X_i$ converges in distribution to a normal distribution.*

$$\frac{S_n - \mathbb{E}[S_n]}{s_n} \xrightarrow{d} \mathcal{N}(0, 1)$$

*Proof.* The proof of Theorem 4.2 applies Lemma 3 to the family of centered random variables

$$\rho_{uv} \triangleq \hat{\mathbf{U}}_{u,p} \hat{\mathbf{U}}_{v,q} \left(\mathbb{Y}_{uv}(\phi^b) - \mathbb{A}_{uv}(\phi^b)\right),$$

which are the $N^2$ terms in the sum forming the numerator of Equation 3 in Section 4.3:

$$\hat{\mathbb{S}}_{pq}(\phi^b) - \mathbb{E}[\hat{\mathbb{S}}_{pq}(\phi^b)] = \sum_{u,v \in [N]^2} \rho_{uv}.$$

The variables $\rho_{uv}$ satisfy $\mathbb{E}[\rho_{uv}] = 0$, and their variance is given by

$$\mathrm{Var}[\rho_{uv}] = \hat{\mathbf{U}}_{u,p}^2 \hat{\mathbf{U}}_{v,q}^2 \, \mathrm{Var}[\mathbb{Y}_{uv}(\phi^b)] = \hat{\mathbf{U}}_{u,p}^2 \hat{\mathbf{U}}_{v,q}^2 \mathbb{A}_{uv}((\phi^b)^2).$$

This last equality follows from the fact that a Poisson process projection $\int_{\mathcal{T}} \phi^b(t) d\mathbb{Y}_{uv}(t)$ can be viewed as a weighted sum of independent infinitesimal Poisson increments $d\mathbb{Y}_{uv}(t) \sim$ Poisson$(\mathbb{A}_{uv}(t)dt)$, each having Poisson variance $\mathbb{A}_{uv}(t)dt$ and weighted by $\phi^b(t)$. These weights get squared in the variance, leading to:

$$\mathrm{Var}\left(\int_{\mathcal{T}} \phi^b(t) d\mathbb{Y}_{uv}(t)\right) = \int_{\mathcal{T}} \phi^b(t)^2 \mathbb{A}_{uv}(t) dt = \mathbb{A}_{uv}((\phi^b)^2).$$

We will verify the Lyapunov condition with $\delta = 1$. Namely, our goal is to show that

$$\frac{1}{s_N^3} \sum_{u,v \in [N]^2} \mathbb{E}[|\rho_{uv}|^3] \to 0 \quad \text{as } N \to \infty,$$

where the cumulated variance in the denominator is defined as

$$s_N^2 \triangleq \sum_{u,v \in [N]^2} \hat{\mathbf{U}}_{u,p}^2 \hat{\mathbf{U}}_{v,q}^2 \, \mathbb{A}_{uv}((\phi^b)^2).$$

To do so, we upper-bound the third absolute moments and lower-bound the cumulated variance.

**Lower bounding the cumulated variance.**    We have that

$$\mathbb{A}_{uv}\left((\phi^b)^2\right) = \int \phi^b(t)^2 \Lambda_{uv}(t)\, dt$$

$$\geq \alpha_N \int \phi^b(t)^2\, dt$$

$$= \alpha_N \cdot \|\phi^b\|_2^2$$

$$= \alpha_N,$$

where the inequality follows by assumption.

Substituting this inequality into the previous equation yields

$$s_N^2 \geq \alpha_N \cdot \sum_{u,v \in [N]^2} \hat{\mathbf{U}}_{u,p}^2 \hat{\mathbf{U}}_{v,q}^2$$

$$= \alpha_N \cdot \left( \sum_{u \in [N]} \hat{\mathbf{U}}_{u,p}^2 \right) \left( \sum_{v \in [N]} \hat{\mathbf{U}}_{v,q}^2 \right)$$

$$= \alpha_N \cdot \|\hat{\mathbf{U}}_{:,p}\|_2^2 \|\hat{\mathbf{U}}_{:,q}\|_2^2$$

$$= \alpha_N,$$

where the final equality holds since the columns of $\hat{\mathbf{U}}$ have unit norm. This shows that the cumulative variance is lower bounded by the factor $\alpha_N$, namely the lower bound on the intensity function.

**Upper bounding the third moment of $\rho_{uv}$.** Now that we have lower bounded the cumulated variance, we need to upper bound the third moment of $\rho_{uv}$. We have

$$\mathbb{E}\left[|\rho_{uv}|^3\right] = |\hat{\mathbf{U}}_{up}|^3 |\hat{\mathbf{U}}_{vq}|^3 \mathbb{E}\left[|\mathbb{Y}_{uv}(\phi^b) - \mathbb{A}_{uv}(\phi^b)|^3\right]$$

By assumption, $|\hat{\mathbf{U}}_{up}|^3 |\hat{\mathbf{U}}_{vq}|^3 \leq (\mu_N/n)^3$, and by the Cauchy-Schwartz inequality, we have

$$\mathbb{E}\left[|\mathbb{Y}_{uv}(\phi^b) - \mathbb{A}_{uv}(\phi^b)|^3\right] = \mathbb{E}\left[|\mathbb{Y}_{uv}(\phi^b) - \mathbb{A}_{uv}(\phi^b)|^1 |\mathbb{Y}_{uv}(\phi^b) - \mathbb{A}_{uv}(\phi^b)|^2\right]$$

$$\leq \sqrt{\underbrace{\mathbb{E}\left[\left(\mathbb{Y}_{uv}(\phi^b) - \mathbb{A}_{uv}(\phi^b)\right)^2\right]}_{m_2} \underbrace{\mathbb{E}\left[\left(\mathbb{Y}_{uv}(\phi^b) - \mathbb{A}_{uv}(\phi^b)\right)^4\right]}_{m_4}}$$

The two factors in the right-hand side are the second and fourth central moments of $\mathbb{Y}_{uv}(\phi^b)$, which we denote as $m_2$ and $m_4$ respectively. These moments relate to the so-called cumulants $\kappa_2$ and $\kappa_4$ of $\mathbb{Y}_{uv}(\phi^b)$, as we have:

$$m_2 = \kappa_2$$

$$m_4 = \kappa_4 + 3\kappa_2^2,$$

where $\kappa_2$ is the second cumulant and $\kappa_4$ is the fourth cumulant of the random variable $\mathbb{Y}_{uv}(\phi^b)$. We will use Campbell's theorem from [22] to express $\kappa_2$ and $\kappa_4$. For the Poisson Processes $\mathbb{Y}_{uv}$ with intensity $\Lambda_{uv}$ and any measurable function $\phi$, the cumulant generating function is given by Campbell's theorem (Equation 3.6 from [22]) by:

$$K(\lambda) = \log(\mathbb{E}\left[\exp\left(\lambda \mathbb{Y}_{uv}(\phi)\right)\right]) = \int \left(e^{\lambda \phi(t)} - 1\right) \Lambda_{uv}(t) dt$$

By expanding $e^{\lambda \phi^b(t)} - 1 = \sum_{k=1}^{\infty} \frac{\lambda^k}{k!} (\phi^b(t))^k$ and applying the linearity of the integral and the fact that $(\phi^b)^k \Lambda_{uv}$ are all compactly supported and continuous (hence integrable), we get:

$$K(\lambda) = \sum_{k=1}^{\infty} \frac{\lambda^k}{k!} \int \phi^b(t)^k \Lambda_{uv}(t) dt$$

By evaluating the second and fourth derivatives of $K(\lambda)$ at $\lambda = 0$, we obtain the second and fourth cumulants

$$\kappa_2 = \Lambda_{uv}\left((\phi^b)^2\right) = \int \phi^b(t)^2 \Lambda_{uv}(t) \, dt, \quad \kappa_4 = \Lambda_{uv}\left((\phi^b)^4\right) = \int \phi^b(t)^4 \Lambda_{uv}(t) \, dt.$$

By assumption, $\mathbf{\Lambda}_{uv}(t) \leq \beta_N$, and since $\phi^b$ are fixed, we have

$$\kappa_2 = \int \phi^b(t)^2 \Lambda_{uv}(t) \, dt \leq \beta_N \int \phi^b(t)^2 \, dt = \beta_N \|\phi^b\|_2^2 = \beta_N,$$

$$\kappa_4 = \int \phi^b(t)^4 \Lambda_{uv}(t) \, dt \leq \beta_N \int \phi^b(t)^4 \, dt \stackrel{\Delta}{=} \eta_N = \mathcal{O}(\beta_N).$$

Therefore

$$\mathbb{E}\left[|\mathbb{Y}_{uv}(\phi^b) - \mathbb{A}_{uv}(\phi^b)|^3\right] = \sqrt{m_2\,m_4} = \sqrt{\kappa_2\left(\kappa_4 + 3\,\kappa_2^2\right)} \leq \sqrt{\beta_N\left(\eta_N + 3\,\beta_N^2\right)} = \mathcal{O}(\beta_N^{3/2}).$$

As a result, we have

$$\mathbb{E}\left[|\rho_{uv}|^3\right] = \mathcal{O}\left\{\left(\frac{\mu_N}{N}\right)^3 \beta_N^{3/2}\right\}.$$

Summing over the $N^2$ terms, we obtain

$$\sum_{u,v\in[N]^2} \mathbb{E}[|\rho_{uv}|^3] = \mathcal{O}\left(\frac{\mu_N^3\beta_N^{3/2}}{N}\right).$$

Dividing this expression by $s_N^3$ gives

$$\frac{1}{s_N^3}\sum_{u,v\in[N]^2} \mathbb{E}[|\rho_{uv}|^3] = \mathcal{O}\left\{\frac{\mu_N^3}{N}\left(\frac{\beta_N}{\alpha_N}\right)^{3/2}\right\} \to 0 \qquad \text{as } N \to \infty$$

which vanishes by assumption.

This shows that the Lyapunov condition is satisfied, and we can apply the Lyapunov CLT (Lemma 3) to conclude that as $N \to \infty$,

$$\frac{\displaystyle\sum_{u,v\in[N]^2} \rho_{uv}}{s_N} = \frac{\displaystyle\sum_{u,v\in[N]^2} \hat{\mathbf{U}}_{u,p}\hat{\mathbf{U}}_{v,q}\left[\mathbb{Y}_{uv}(\phi^b) - \mathbb{A}_{uv}(\phi^b)\right]}{\sqrt{\displaystyle\sum_{u,v\in[N]^2} \hat{\mathbf{U}}_{u,p}^2\hat{\mathbf{U}}_{v,q}^2\mathbb{A}_{uv}\left((\phi^b)^2\right)}} \xrightarrow{d} \mathcal{N}(0,1).$$

$\square$

# D   Experimental Setup

## D.1   Synthetic Data Generation

We generate two types of synthetic networks to evaluate our methods: the Erdös–Rényi (ER) blocks model and a Dynamic Stochastic Block Model (DSBM). Their time-varying intensity functions are shown in Figure 4.

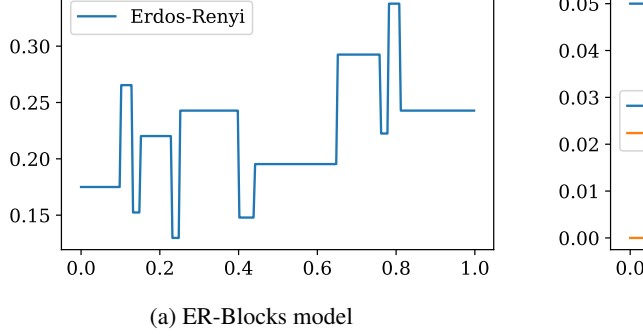

(a) ER-Blocks model

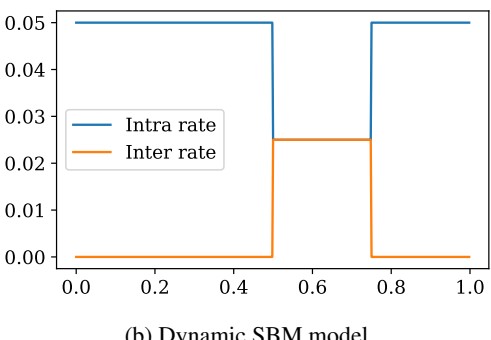

(b) Dynamic SBM model

Figure 4: Intensity functions for the synthetic network models. (a) ER-blocks uses a piecewise-constant intensity with abrupt jumps. (b) DSBM distinguishes intra-community (blue) and inter-community (orange) intensities, with a mid-experiment perturbation.

### D.1.1 Erdös–Rényi (ER) Blocks Model

In this model, the intensity between every node pair is the same, and defined as the following piecewise-constant function:

$$\Lambda_{uv}(t) \;=\; \sum_{k=1}^{K} h_k \, K\big(t - t_k\big), \qquad K(x) = \frac{1 + \mathrm{sign}(x)}{2},$$

with

$$\{t_k\}_{k=1}^{K} = \{0.10,\, 0.13,\, 0.15,\, 0.23,\, 0.25,\, 0.40,\, 0.44,\, 0.65,\, 0.76,\, 0.78,\, 0.81\},$$

$$\{h_k\}_{k=1}^{K} = \{4,\, -5,\, 3,\, -4,\, 5,\, -4.2,\, 2.1,\, 4.3,\, -3.1,\, 5.1,\, -4.2\}.$$

This model, adapted from the synthetic example from [15], simulates a network with a single community, where the interaction intensity only depends on time, and not on other latent factors such as community assignments.

### D.1.2 Dynamic Stochastic Block Model (DSBM)

In this model, the nodes are partitioned into two communities $\mathcal{C}_1$ and $\mathcal{C}_2$. This time the intensity function varies depending on whether the node pair belongs to the same community (genering intra-community interactions) or to different communities (inter-community interactions):

$$\Lambda_{uv}(t) = \begin{cases} \lambda_{\mathrm{intra}}(t), & u, v \in \mathcal{C}_1 \text{ or } u, v \in \mathcal{C}_2, \\ \lambda_{\mathrm{inter}}(t), & \text{otherwise.} \end{cases}$$

As shown in Figure 4b, both the intra and inter community intensities are piecewise-constant functions. The intra-community intensity is set much higher than the inter-community intensity except on an interval $[0.5, 0.75]$ where both intensities are equal. This model simulates the temporary fusion of two communities into a single one.

## D.2 Hyperparameter selection

We now give more details on the methods used in the intensity estimation experiment and the associated hyperparameters. We experimented with various parameters for the different method, and selected the ones which yielded the lowest MISE.

Table 1: Hyperparameter selection for different methods

| Method | Parameter | ER-blocks dataset | SBM dataset |
|---|---|---|---|
| IPP-KDE | Bandwidth | 0.005 | 0.05 |
| IPP-Hist | Number of bins ($M$) | 128 | 64 |
| ANIE (ours) | Resolution level ($J$) | 8 | 6 |
| | Significance level ($\alpha$) | 0.05 | 0.05 |

## D.3 Resources

**Hardware used for the experiments** All the experiments we run on a MacBook Air with an Apple M1 chip with 8 CPU cores and 8GB of RAM.

**Fitting time of ANIE** We report the fitting time of ANIE vs the number of nodes in Figure 5 for different levels for the maximum resolution $J$ of the Haar basis (which modulates the size of the orthonormal basis). We observe that the fitting time of ANIE is quadratic in the number of nodes, and scales exponentially with the number of levels. This underlines a limitation: in order to capture fine grained change, the number of levels $J$ must be large, which leads to an exponentially large number of coefficients to process. However, some optimizations could be made such as parallelizing the computation of the coefficients, or using a more efficient algorithm to compute the truncated SVD.

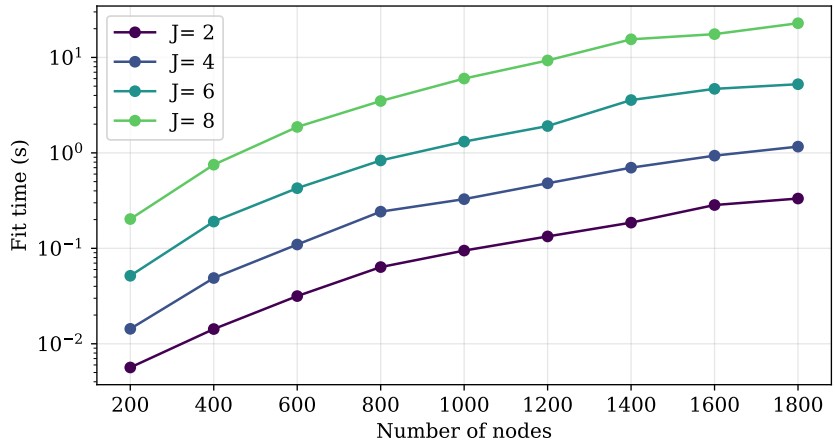

Figure 5: Fitting time of ANIE vs number of nodes for different values of $J$.

# E  Effect of the Hyperparameters of ANIE

In order to better illustrate the effect of the resolution $J$ on the estimation error, we ran the ANIE method on a simplified SBM dataset with different values. In this simplified setting, we parameterized the model such that a resolution $J = 2$ is sufficient to capture the intensity function. We then ran the ANIE method with different values of $J$ and compared the estimation error of the linear and thresholded estimators.

**Effect of the number of levels** $J$    Figure 6 shows the effect of the number of levels $J$ on the estimation error. We observe that the linear estimator performs well for small values of $J$, but its performance degrades as $J$ increases. In contrast, the thresholded estimator maintains a low estimation error across all values of $J$, demonstrating its robustness to overfitting.

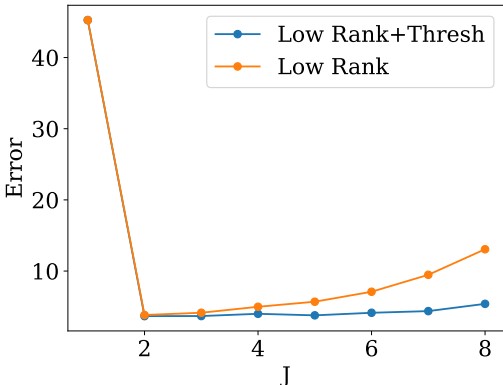

Figure 6: Estimation error vs number of levels for the linear and the thresholded estimator

# F  LAD Implementation

We use the publicly available implementation https://github.com/shenyangHuang/LAD to compare our method with LAD. We use the default parameters of the implementation.

# G   Case Studies: London Bike Dataset and Enron email network

Both dataset described below may be viewed as temporal networks, where continuous interactions represent trips between bike stations in London, or email exchanges between Enron employees. For both the datasets, we plot two discrete wavelet scaleograms which we here refer to as the naive and reconstructed scaleograms, by computing for each level $j$, the Frobenius norm of either the naive coefficient estimates $\|\mathbb{Y}(\psi_{jk})\|_F$ or the empirical affinity coefficients $\|\hat{\mathbb{S}}(\psi_{jk})\|_F$. It can be shown that the latter corresponds to the Frobenius norm of a low-rank reconstruction of the naive coefficients. Specifically, using the orthonormality of $\hat{\mathbf{U}}$, we have $\|\hat{\mathbb{S}}(\psi_{jk})\|_F = \|\hat{\mathbf{U}}\hat{\mathbf{U}}^T\mathbb{Y}(\psi_{jk})\hat{\mathbf{U}}\hat{\mathbf{U}}^T\|_F$. In both cases, we typically observe that the wavelet power of the reconstructed affinities between latent factors is more concentrated in low-frequency bands, while the naive per-edge estimates $\|\mathbb{Y}(\psi_{jk})\|_F$ exhibit more energy in the high-frequency bands.

## G.1   London Bike Dataset

The London Bike dataset, published by Transport for London [1], has an inherent dynamic network structure which has been previously studied for instance in [36, 41]. We consider a week of data, from 1st to 8th of May 2017, and mark each starting trip from docking station $u$ to docking station $v$ at time $t$ an instantaneous event. This results in 219515 interactions between 780 nodes (bike stations). As shown on Figure 7, plotting the naive and reconstructed wavelet scaleograms for this dataset allows us to identify periods and time scales during which significant structural changes occur, and to contrast these with changes that are merely due to fluctuations in exchange intensity between individual pairs of bike stations. Moreover, the estimate obtained using ANIE enables us to visualize the bike stations in a low-dimensional space using a t-SNE plot, where stations that connect to similar neighborhoods at similar times appear close together. Coloring the bike stations by London borough reveals that stations from some boroughs, such as Tower Hamlets or Newham, tend to cluster tightly. In contrast, stations from boroughs like Westminster, Kensington and Chelsea, or Hammersmith and Fulham are more dispersed, indicating greater diversity in their patterns of use.

## G.2   Enron Email Dataset

For the Enron dataset, as shown on Figure 8, we find that both the naive and reconstructed scaleograms capture changes in 2001, which marked the buildup to the company's bankruptcy.

These changes are concentrated in a specific frequency band, illustrating the ability of our method to identify the time scale at which structural shifts occur. Additionally, we observe that, in the raw scaleogram (Figure 8b), measuring change purely at the edge-level, events following 2001 are associated with changes across a wider range of frequency bands. In contrast, the reconstructed, denoised scaleogram (Figure 8c) shows less variation during this later period, suggesting that many of the post-2001 edge-level fluctuations do not correspond to substantial structural changes in the network, or at least not to the same extent as during the major 2001 events.

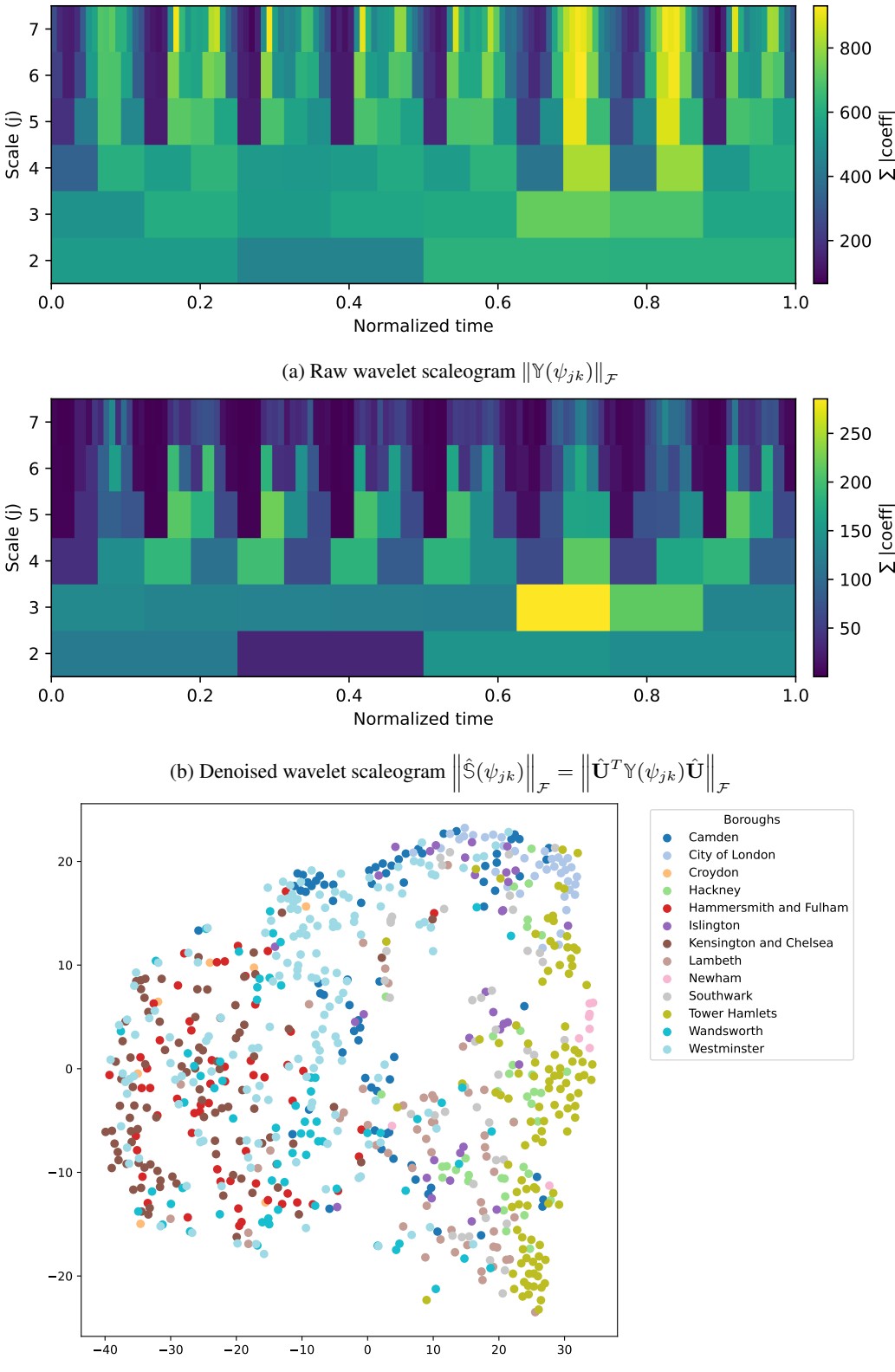

(a) Raw wavelet scaleogram $\left\|\mathbb{Y}(\psi_{jk})\right\|_{\mathcal{F}}$

(b) Denoised wavelet scaleogram $\left\|\hat{\mathbb{S}}(\psi_{jk})\right\|_{\mathcal{F}} = \left\|\hat{\mathbf{U}}^T \mathbb{Y}(\psi_{jk})\hat{\mathbf{U}}\right\|_{\mathcal{F}}$

(c) t-SNE embedding of the rows estimated subspace matrix $\hat{\mathbf{U}}$ colored by the boroughs of London

Figure 7: Visualization of the ANIE results on the the London Bike dataset: raw and denoised wavelet scaleograms, and t-SNE embedding of the estimated latent factors.

| Event | Date | Description |
|-------|------|-------------|
| 1 | Nov 1999 | Enron launched |
| 2 | Feb 2001 | Jeffrey Skilling takes over as CEO |
| 3 | 14 Aug 2001 | Kenneth Lay takes over as CEO after Skilling resigns |
| 4 | 9 Nov 2001 | Enron restates 3rd quarter earnings revealing massive losses |
| 5 | 29 Nov 2001 | Dynegy deal collapses, ending Enron's last hope for rescue |
| 6 | 10 Jan 2002 | Department of Justice confirms criminal investigation begun |
| 7 | 23 Jan 2002 | Kenneth Lay resigns as CEO amid investigations |
| 8 | 4 Feb 2002 | Lay implicated in plot to inflate profits and hide losses |
| 9 | 24 Apr 2002 | U.S. House passes accounting reform package in response to Enron scandal |

(a) Timeline of key events in the Enron scandal.

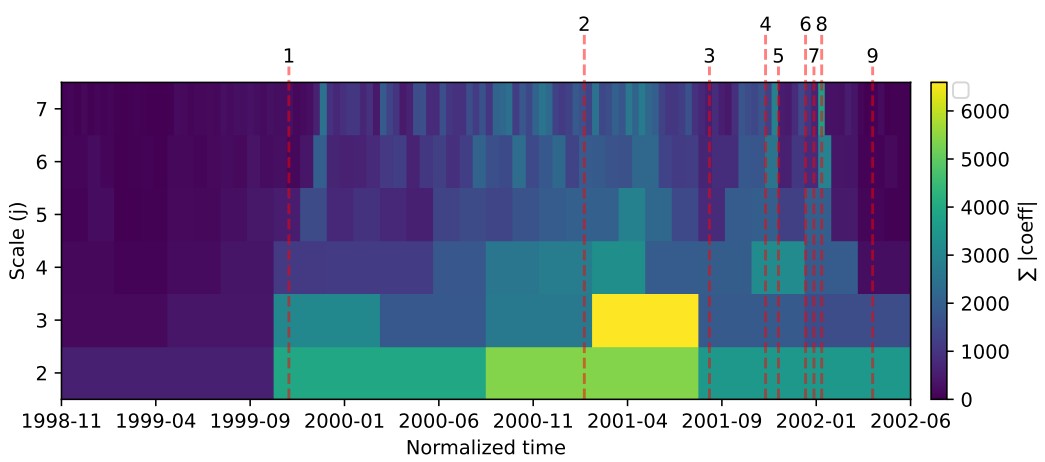

(b) Raw wavelet scaleogram $\|\mathbb{Y}(\psi_{jk})\|_{\mathcal{F}}$.

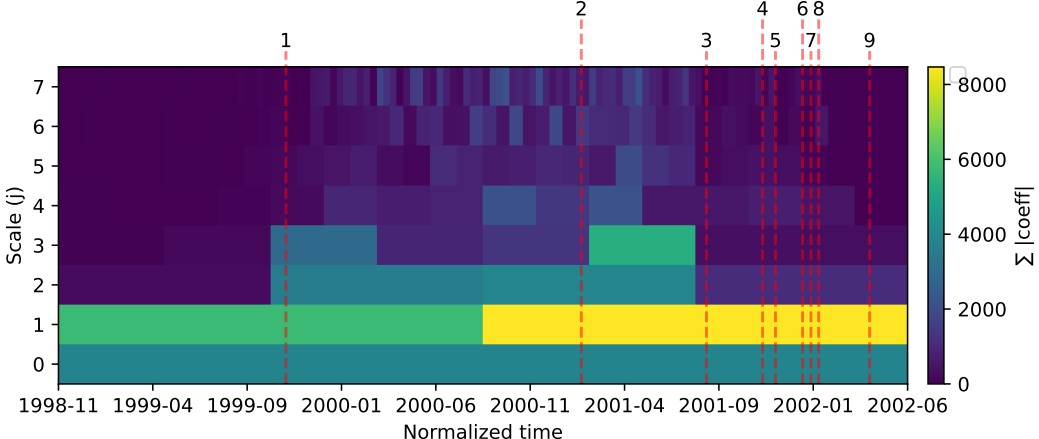

(c) Denoised wavelet scaleogram $\left\|\hat{\mathbb{S}}(\psi_{jk})\right\|_{\mathcal{F}} = \left\|\hat{\mathbf{U}}^T \mathbb{Y}(\psi_{jk}) \hat{\mathbf{U}}\right\|_{\mathcal{F}}$.

Figure 8: Discrete wavelet scaleogram analysis of the Enron email dataset. The top table lists the key events annotated in the scalograms below.

