# OpenReview forum: "Multiresolution Analysis and Statistical Thresholding on Dynamic Networks"
_NeurIPS.cc/2025/Conference — NeurIPS 2025 poster_

### Official Review · Reviewer_HVGn · 2025-07-01

**Clarity:** 3
**Significance:** 2
**Originality:** 3
**Rating:** 5
**Confidence:** 2

**Summary:**

This paper proposes a novel method for change detection in evolving graphs. The main goal of the approach is to provide the flexibility to be able to capture both highly localized changes, which might correspond to bursts in time, as well as broader changes. The authors introduce a time-continuous model that they call COSIP (Common Subspace Independent Processes), which is based on a shared subspace and a time-dependent affinity function, which captures the time-dependent changes. Given this theoretical model, detecting changes in the network structure can then be reduced to extracting the temporal features of the matrix-valued affinity function. The authors thus provide a practical algorithm for estimating the two components of their model from data, and provide theoretical consistency guarantees for their estimators. One nice feature of this model is that it is quite flexible and, for example, the basis can be chosen depending on the specific scenario. Finally, the authors also show experimental evidence that their model produces meaningful results in both synthetic and real-world scenarios, by comparing their resulting method against different baselines.

**Questions:**

1. As a non-expert in this domain, I was surprised to see that, as the authors mentioned, “Note that, IPP-Hist is equivalent to Anie with Haar wavelet without thresholding”. I feel that this should have been mentioned earlier on, when describing the theoretical model, which is presented without any prior background. The authors do mention that the model extends the COSIE model [3] but do not reference IPP-Hist
2. Since the model provided by the authors is an extension of the COSIE model, I would expect a comparison against that baseline in the experimental section. Why did the authors choose not to include such a comparison?
3. I’m also wondering to what extent the assumption that the basis is shared (time-independent) in the model is realistic? I would imagine that in some datasets, there can be slow time-dependent changes that alter the entire global structure of the graph and those cannot be meaningfully captured by a universal basis, unless the size of the basis is very large. Have the authors witnessed such a scenario?

**Ethical Concerns:**

["NO or VERY MINOR ethics concerns only"]

**Final Justification:**

The authors have addressed the main issues raised in my review in their rebuttal, and I'm happy to recommend this paper for acceptance.

**Limitations:**

Yes, the authors discuss the limitations of the proposed approach in Section 6 of the paper. They mention that one limitation is the method's reliance on variance estimation for Z-score computation, which may compromise thresholding robustness, especially at high resolutions. While this highlights one particular limitation, I am also wondering about the generality and applicability of the proposed theoretical model, which would be worth discussing as well.

**Paper Formatting Concerns:**

The paper is well-formatted. I do not see any issuses.

**Quality:**

3

**Strengths And Weaknesses:**

First, I should say that I am not an expert in this domain and therefore it is not easy for me to judge the novelty or scope of the contribution of this paper. From my perspective, the main strength of the method is the rigorous model behind the approach, which, as the authors highlight, combines robustness against high-frequency noise, together with the ability to meaningfully capture even rapid temporal changes. In addition, the flexibility of the model (with the ability to choose the basis) and the theoretical analysis, including consistency guarantees are all strengths of the proposed approach.

1. The main weakness is probably the relatively weak experimental evaluation and inconsistent choice of baselines. This includes:
The key comparisons in Figure 2 seem to only be given in a qualitative manner. There is no quantitative evaluation against IPP-Hist and IPP-KDE which are the two closest baselines.
2. The evaluation on real data (Section 5.2) is given against the Laplacian Anomaly Detection (LAD) baseline and not against IPP-Hist baselines. This is somewhat strange, since LAD is now over 15 years old, so comparing *only* against such a baseline is very limiting.


I also have questions about the novelty and relation to prior work mentioned in the Questions section. Generally speaking, the lack of consistency of the baselines against which the method, and the exposition which (at least to a non-expert like myself) does not make it entirely clear which exact components are new and which ones are borrowed from existing works, be a limitation of the proposed approach.

---

> ### Author Rebuttal · Authors · 2025-07-30
>
> We thank reviewer HVGn for their time and detailed comments. We were glad to hear the reviewer appreciated the rigorous model, the flexibility of the approach, and the presence of theoretical guarantees.
>
> > The key comparisons in Figure 2 seem to only be given in a qualitative manner. There is no quantitative evaluation against IPP-Hist and IPP-KDE which are the two closest baselines.
>
>
> We would like to point out that Figs. 2(e) and 2(h) provide a quantitative evaluation of intensity estimation. In these plots, our method is compared against the described baselines using the Mean Integrated Squared Error (MISE) metric, across varying numbers of nodes. These figures are intended to complement the visual insights from Figs. 2(a)–2(f) with a quantitative comparison.
>
> > The evaluation on real data (Section 5.2) is given against the Laplacian Anomaly Detection (LAD) baseline and not against IPP-Hist baselines.
>
> The IPP-Hist and IPP-KDE baselines where not included in the change detection experiment as these methods were not designed as change detection method but as intensity estimation and dimensionality reduction method. As a result we decided to compare with the change point detection method closest to our work, namely LAD.
>
> > This is somewhat strange, since LAD is now over 15 years old, so comparing only against such a baseline is very limiting.
>
> We would like to clarify that the Laplacian Anomaly Detection (LAD) method used in our work was first introduced at KDD 2020 [1], with an extended version published in ACM TKDD in 2023 [2]. It appears there may be some confusion with a different method that uses the same acronym. In the revised version of the paper, we have incorporated the reviewer's suggestion, which also addresses a related question from reviewer AekB, and strengthened our evaluation of change detection by including an additional comparison with the TensorSplat method [3].
>
> > As a non-expert in this domain, I was surprised to see that, as the authors mentioned, “Note that, IPP-Hist is equivalent to Anie with Haar wavelet without thresholding”. I feel that this should have been mentioned earlier on, when describing the theoretical model, which is presented without any prior background. The authors do mention that the model extends the COSIE model [3] but do not reference IPP-Hist.
>
> Thank you for your observation. To clarify, the statement refers to an overlap between very special cases of the IPP and ANIE methods. More precisely, it is established in the literature that estimating the intensity of a point process using the Haar wavelet basis at resolution level $J$ is exactly equivalent to histogram estimation with $2^J$ equal-width bins (each of width $2^{-J}$). However, this equivalence is limited to this particular configuration and does not mean that ANIE is generally a special case of IPP-Hist.
>
> In fact, there are fundamental differences between the two methods. Notably, ANIE is a non-linear approach because it incorporates adaptive thresholding on the wavelet coefficients, whereas IPP-Hist does not. We will make sure to clarify this relationship in the manuscript to avoid any potential confusion.
>
> > Since the model provided by the authors is an extension of the COSIE model, I would expect a comparison against that baseline in the experimental section. Why did the authors choose not to include such a comparison?
>
> Thank you for your question. We would like to clarify that the COSIP model, like COSIE, is not itself an estimation method, but rather a statistical modelused to analyze our proposed ANIE method. Neither COSIP nor COSIE provides an algorithm for intensity estimation.
>
> The main similarity between COSIE and COSIP is that both assume a single subspace structure across the entire observation interval. However, COSIE is specifically designed for collections of discrete-time graphs defined on the same set of nodes, while our work addresses the continuous-time setting, which COSIE does not cover.
>
> Additionally, the MASE method introduced in the COSIE paper can be extended only to discrete-time graphs and does not directly extend to the continuous-time context that is the focus of our work. For these reasons, we did not include COSIE/MASE in our quantitative comparisons. We will clarify this distinction in the revised manuscript.
>
> > I’m also wondering to what extent the assumption that the basis is shared (time-independent) in the model is realistic? I would imagine that in some datasets, there can be slow time-dependent changes that alter the entire global structure of the graph and those cannot be meaningfully captured by a universal basis, unless the size of the basis is very large. Have the authors witnessed such a scenario?
>
> *Note: Joint Response with reviewer AekB*
>
> The assumption of a global low-dimensional subspace that is constant across time is a common one in the dynamic network modelling literature (see e.g. [4]–[9]).
>
> While it may seem restrictive, one should note that the dimension of the subspace $U$ may be larger than the rank of the intensity matrix at any given time, allowing for dynamic community structures. There's a sense in which the rank $D$ represents the diversity of behaviours of nodes across time.
>
> Consider the example of a two-community dynamic stochastic block model on the time interval $[0,T)$ with constant intra-community intensity $a$ and inter-community intensity $b$. This example is similar to those given in [6], [7], and [8]. Suppose that at time $T/2$, half of community one moves to community two. This model can be represented as $\\mathbf{\\Lambda}(t) = \\mathbf{U}\\mathbf{S}(t)\\mathbf{U}^\\top$ where:
>
> 3. The $i$th row of $\\mathbf{U}$ is a one-hot vector with a one in position 1 if node $i$ belongs to community one, a one in position 2 if node $i$ starts in community one and moves to community two, and a one in position 3 if node $i$ is in community two.
>
> 4. The matrix $\\mathbf{S}(t)$ is given by:
>  $$ \\mathbf{S}(t) = \\begin{pmatrix} a(t) & a(t) & b(t) \\\\ a(t) & a(t) & b(t) \\\\ b(t) & b(t) & a(t) \\end{pmatrix}, \\qquad t \\in [0,T/2) $$
>    $$ \\mathbf{S}(t) = \\begin{pmatrix} a(t) & b(t) & b(t) \\\\ b(t) & a(t) & a(t) \\\\ b(t) & a(t) & a(t) \\end{pmatrix}, \\qquad t \\in [T/2,T) $$
> At any given point in time, the intensity matrix has rank 2, and could be represented in the model on (different) 2-dimensional subspaces, yet we can represent them using a *global* subspace of dimension 3.
>
> ***References***
>
>
> [1] Huang, Shenyang, Yasmeen Hitti, Guillaume Rabusseau, and Reihaneh Rabbany. “Laplacian Change Point Detection for Dynamic Graphs.” Proceedings of the 26th ACM SIGKDD International Conference on Knowledge Discovery & Data Mining (New York, NY, USA), KDD ’20, Association for Computing Machinery, août 2020, 349–58.
>
> [2] Huang, Shenyang, Samy Coulombe, Yasmeen Hitti, Reihaneh Rabbany, and Guillaume Rabusseau. “Laplacian Change Point Detection for Single and Multi-View Dynamic Graphs.” ACM Transactions on Knowledge Discovery from Data 18, no. 3 (2024): 1–32. Yasmeen Hitti, Guillaume Rabusseau, and Reihaneh Rabbany.
>
> [3] Koutra, D., Papalexakis, E. E., & Faloutsos, C. (2012). TensorSplat: Spotting Latent Anomalies in Time. Proceedings of the 16th Panhellenic Conference on Informatics, 144–149.
>
> [4] Jones, A., & Rubin-Delanchy, P. (2020). The multilayer random dot product graph. *arXiv preprint arXiv:2007.10455*.
>
> [5] Zhang, X., Xue, S., & Zhu, J. (2020, November). A flexible latent space model for multilayer networks. In *International Conference on Machine Learning* (pp. 11288–11297). PMLR.
>
> [6] Arroyo, J., Athreya, A., Cape, J., Chen, G., Priebe, C. E., & Vogelstein, J. T. (2021). Inference for multiple heterogeneous networks with a common invariant subspace. *Journal of Machine Learning Research*, *22*(142), 1–49.
>
> [7] Gallagher, I., Jones, A., & Rubin-Delanchy, P. (2021). Spectral embedding for dynamic networks with stability guarantees. *Advances in Neural Information Processing Systems*, *34*, 10158–10170.
>
> [8] Jing, B. Y., Li, T., Lyu, Z., & Xia, D. (2021). Community detection on mixture multilayer networks via regularized tensor decomposition. *The Annals of Statistics*, *49*(6), 3181–3205.
>
> [9] Agterberg, J., Lubberts, Z., & Arroyo, J. (2025). Joint spectral clustering in multilayer degree-corrected stochastic blockmodels. *Journal of the American Statistical Association*, (just-accepted), 1–23.

---

> > ### Comment · Reviewer_HVGn · 2025-08-04
> >
> > I would like to thank the reviewers for their thorough answer. The rebuttal addresses the main questions that I had about this work. I am therefore happy to recommend this paper for acceptance. I would like to ask the authors to
> >
> > 1. Incorporate all of the promised additional results, such as a comparison to the TensorSplat baseline.
> > 2. Add a more thorough discussion regarding the applicability of their approach, and possible *limitations* of the theoretical model in this work. I think highlighting the limitations of the proposed approach will only help to increase its impact.
> > 3. If possible, release the implementation of their method, ensuring the full reproducibility of the presented results, in order to allow follow-up work.

---

> > > ### Author Response · Authors · 2025-08-06
> > >
> > > We thank reviewer HVGn once again for their review and suggestions. We will carefully incorporate these into the revision, along with a link to the GitHub repository containing the implementation and code to reproduce the results.

---

### Official Review · Reviewer_Kwi7 · 2025-07-02

**Clarity:** 3
**Significance:** 3
**Originality:** 3
**Rating:** 5
**Confidence:** 3

**Summary:**

This paper proposes a point-process based dynamic network modeling approach, which models the temporal network at the scale of nodes. Moreover, the wavelet signal decomposition is transferred to deal with the local changes in the intensity functions. The theoretical analysis is provided to demonstrate the consistentcy of subspace estimation and experiments as well as case studies are conducted to evluate the performance of the proposed method.

**Questions:**

1）Is  the proposed method capable of online change-point detection?
2)  In implementing the comparison method LDA,  does you perform the optimal-parameter search?
3)  It is hard to see the degree of agreement between the predicted change-point and the ground truth  in Fig.3(b).

**Ethical Concerns:**

["NO or VERY MINOR ethics concerns only"]

**Final Justification:**

The paper is well-written and demonstrates technical novelty. All my concerns have been adequately addressed, and I recommend acceptance.

**Limitations:**

yes

**Paper Formatting Concerns:**

no formatting issues.

**Quality:**

3

**Strengths And Weaknesses:**

Strengths:
1. The proposed method shows the adaptability to the time resolution of dynamic network and offers an subspace decomposition approach for low-rank computing.
2. Wavelet is introduced to network dynamic analysis.
3. The experiments on synthetic data and real data demonstrate the effectiveness of the proposed method, especially on precisely change-dynamics fitting.

Weaknesses:
1. The evaluation is limited to single change-point synthetic networks and one real-world networks. More datasets e.g., Enron email networks and Reddit that  include multiple change-points will improve the comprehensive performance evaluation of the proposed method.
2. Quantitative measuring the success of hitting the change points and how much the method accurately identify the change points for multiple change-point case are required, besides the visualization of the results, e.g., Adjusted F1.
3. Incorporating additional baseline methods will strengthen the comparative analysis, such as the representation learning/network comparsion based methods and time window base methods (as mentioned in the paper).

---

> ### Author Rebuttal · Authors · 2025-07-30
>
> We thank reviewer Kwi7 for their comments and are pleased that they appreciated the novelty of applying low-rank methods and wavelet analysis to dynamic networks. Below is a point-by-point response to the reviewer's questions.
>
> > Quantitative measuring the success of hitting the change points and how much the method accurately identify the change points for multiple change-point case are required, besides the visualization of the results, e.g., Adjusted F1.
>
> We thank the reviewer for raising this point. Given that the considere UCI dataset contains only a limited number of two ground truth change points, quantitative metrics such as Adjusted F1 may not provide substantially more insight beyond the visualizations already included.
>
> > Is the proposed method capable of online change-point detection?
>
> This is a good question. The ANIE method proposed in this paper is inherently offline, as it relies on analyzing a complete history of interactions to identify latent change points in network structure. However, extending ANIE to support online change-point detection represents an interesting and valuable direction for future research. Such an extension could for instance incrementally update the subspace estimates as new data becomes available, and apply multiresolution analysis to incoming batches of newly observed interactions to achieve online change detection.
>
> > In implementing the comparison method LDA, does you perform the optimal-parameter search?
>
> We assume the reviewer is referring to the Laplacian Anomaly Detection (LAD) method, rather than LDA. For our comparison with LAD, we used the authors’ publicly available implementation on GitHub. To ensure fairness and reproducibility, we followed their dataset preprocessing steps and used the hyperparameters provided by the authors, without performing additional tuning. We would also like to note that hyperparameter optimization in this setting is particularly challenging, as change-point detection performance is difficult to quantify. According to the metrics reported in the LAD papers, their method achieves a 100% hit rate on the dataset, with the metric taking only three possible values (0%, 50%, or 100%). This discreteness makes it unsuitable as an objective for hyperparameter selection. Nevertheless, we verified our experimental results by reproducing the anomaly score profile presented in their original paper, and we used this verified result as the baseline for comparison.
>
> > It is hard to see the degree of agreement between the predicted change-point and the ground truth in Fig.3(b).
>
> We agree that overlaying the signals from each level makes it difficult to visualize the ground-truth change-points. First, we would like to clarify what is shown in Fig. 3(b).
>
> - For each resolution level $j$, we compute a discrete time series $E_j(k)$ by taking the sum of absolute values of the empirical affinity matrix $\hat{S}(\phi_{jk})$, which aggregates the total magnitude of change across all pairs of latent factors on the dyadic interval $I_{j,k} = [2^{-j}k, 2^{-j}(k+1)]$:
>
>   $$
>   E_j(k) = \lVert \hat{S}(\phi_{jk}) \rVert_{1}.
>   $$
>
> - We then construct a continuous, piecewise-constant function $E_j(t)$ by setting it equal to $E_j(k)$ over each interval $I_{j,k}$, so that
>
>   $$
>   E_j(t) = E_j(k) \quad \text{for } t \in I_{j,k}.
>   $$
>
>   Each line in the plot in Fig.3(b) corresponds to one such function $E_j(t)$.
>
> In the revised version of the paper, we will clarify the explanation of the plot as described above. Additionally, we will improve Fig. 3(b) by splitting it into eight separate subplots, each representing a different resolution level. This will make it easier to see that some levels are more affected by the ground-truth change points than others. Finally, we will add an aggregate plot showing the sum across all resolution levels, namely $E(t) = \sum_j E_j(t)$, which provides a one-dimensional representation of the amount change over time.

---

> > ### Comment · Reviewer_Kwi7 · 2025-08-04
> >
> > Thank you for your detailed response. My concerns have been adequately addressed, and I have updated my scoring accordingly.

---

### Official Review · Reviewer_AekB · 2025-07-03

**Clarity:** 2
**Significance:** 3
**Originality:** 2
**Rating:** 4
**Confidence:** 3

**Summary:**

This paper proposes ANIE (Adaptive Network Intensity Estimation), a multi-resolution framework designed to detect changes in dynamic networks across multiple temporal resolutions, including both rapid and gradual changes. The method is tested on synthetic and real-world datasets and shows improved performance over selected baselines.

**Questions:**

Can the method be extended to settings where the cross-sectional structure $U$ also evolves over time, such as dynamic community structures?

Are there any theoretical guarantees or approximation-based discussions for controlling false alarm or miss detection probabilities under the multiple testing framework?

In addition to the baseline LAD method used for comparison, there appear to be other change-detection approaches, such as the statistics used in Ref [43] and related works; comparison with these methods—both empirically and conceptually—would strengthen the paper from a change-detection perspective.

Line 240 typo: should be "conservative."

**Ethical Concerns:**

["NO or VERY MINOR ethics concerns only"]

**Final Justification:**

I have increased my score accordingly, as the authors' response addressed my earlier concern about the limited representation of the fixed cross-sectional structure.

**Limitations:**

yes

**Quality:**

3

**Strengths And Weaknesses:**

Strength:
ANIE effectively captures network changes at multiple temporal scales and demonstrates good empirical performance across diverse datasets.

Weakness:
The method assumes a fixed cross-sectional structure $U$, which may limit its applicability in real scenarios, and lacks discussion of false alarm or miss detection probabilities under multiple testing.

---

> ### Author Rebuttal · Authors · 2025-07-30
>
> We thank reviewer AekB for taking the time to review our paper. We hereby provide a point-by-point response to the questions.
>
> > Can the method be extended to settings where the cross-sectional structure also evolves over time, such as dynamic community structures?
>
> *Note: Joint Response with reviewer HVGn.*
>
> The assumption of a global low-dimensional subspace that is constant across time is a common one in the dynamic network modelling literature (see e.g. [1]–[6]).
>
> While it may seem restrictive, one should note that the dimension of the subspace $U$ may be larger than the rank of the intensity matrix at any given time, allowing for dynamic community structures. There's a sense in which the rank $D$ represents the diversity of behaviours of nodes across time.
>
> Consider the example of a two-community dynamic stochastic block model on the time interval $[0, T)$ with constant intra-community intensity $a$ and inter-community intensity $b$. This example is similar to those given in [2], [3], and [4]. Suppose that at time $T/2$, half of community one moves to community two. This model can be represented as $\\mathbf{\\Lambda}(t) = \\mathbf{U}\\mathbf{S}(t)\\mathbf{U}^\\top$ where:
>
> 1. The $i$th row of $\\mathbf{U}$ is a one-hot vector with a one in position 1 if node $i$ belongs to community one, a one in position 2 if node $i$ starts in community one and moves to community two, and a one in position 3 if node $i$ is in community two.
>
> 2. The matrix $\\mathbf{S}(t)$ is given by:
>
>    $$ \\mathbf{S}(t) = \\begin{pmatrix} a(t) & a(t) & b(t) \\\\ a(t) & a(t) & b(t) \\\\ b(t) & b(t) & a(t) \\end{pmatrix}, \\qquad t \\in [0, T/2) $$
>
>    $$ \\mathbf{S}(t) = \\begin{pmatrix} a(t) & b(t) & b(t) \\\\ b(t) & a(t) & a(t) \\\\ b(t) & a(t) & a(t) \\end{pmatrix}, \\qquad t \\in [T/2, T) $$
>
> At any given point in time, the intensity matrix has rank 2, and could be represented in the model on (different) 2-dimensional subspaces, yet we can represent them using a *global* subspace of dimension 3.
>
>
> > Are there any theoretical guarantees or approximation‑based discussions for controlling false alarm or miss detection probabilities under the multiple testing framework?
>
> We thank the reviewer for this important question. We hereby clarify the meaning of false alarms or miss detections in our setting. We remind that following the subspace projection step, we have for each pair of latent factors $p,q$, scale $j$ and location $k$, a hypothesis
> $\mathcal{H}_{j,k}^{p,q} =\text{``} \mathbf{S}\_{p,q}(\phi\_{jk}) = 0 \text{''}$, which indicates that the observed affinity coefficient may be attributed to noise rather than structural change between latent factors $p$ and $q$.
>
> In this context, a false alarm (Type I error) occurs when  $ \\mathcal{H}\_{j,k}^{p,q} $ is true but wrongly rejected, i.e. a noise fluctuation is misinterpreted as a genuine change. This manifests as spurious artefacts in the reconstructed affinity signal.
>
> A miss detection (Type II error) occurs when  $ \\mathcal{H}\_{j,k}^{p,q} $ is false but not rejected, i.e. a real change goes undetected. This leads to missing important features (change points) in the signal. We summarize the two types of error in the following table.
>
> |  $ \\mathcal{H}_{j,k}^{p,q} $ | False (no change) | True (change) |
> | :---- | :---- | :---- |
> | Reject | Correct | Type I error |
> | Accept | Type II error | Correct |
>
> The false positive rate (false alarm rate) is the probability of a Type I error; the false negative rate (miss detection rate) is the probability of a Type II error. In classical single‑hypothesis testing, one sets a significance level  $ \\alpha $ and chooses a threshold  $ \\tau_\\alpha $ such that $ P\\bigl(T\\>\\tau_\\alpha \\mid \\mathcal{H}_0\\bigr) \\le \\alpha $.
>
> In our problem there are  $ D\\times D\\times(2^J-1) $ such hypotheses. We aim to control the total number of Type I (false alarms) and Type II (misses) errors. Two standard quantities that we can control are
>
> * FWER (Family‑Wise Error Rate): the probability of any Type I error among all tests. By testing each hypothesis at level  $ \\alpha/M $ (with  $ M $ total tests) instead of $\alpha$ directly, a union bound gives
>
>    $$\\mathrm{FWER} \\le M\\cdot\\frac{\\alpha}{M} = \\alpha.$$
>
>   This method often referred to as the Bonferroni correction [7] is known to be conservative and reduce detection power. In our case this manifests through some structural changes being ignored in the detection.  This is why, as mentioned in the paper, we opted for the FDR method.
>
> * FDR (False Discovery Rate): the expected proportion of false alarms among all rejections. Let  $ R $ be the number of rejected hypotheses and  $ V $ the number of false rejections. Then, a transformation of the $p$-values known as the Benjamini–Hochberg procedure [8] ensures that
>    $$\\mathrm{FDR} = \\mathbb{E}\\Bigl[\\frac{V}{R\\vee1}\\Bigr] \\le \\alpha$$
>   This approach allows more rejections than FWER control, typically leading to greater statistical power. Since it produced better empirical results in our setting, it is the method we adopted.
>
> While family-wise error rate (FWER) and false discovery rate (FDR) provide theoretical guarantees for controlling Type I errors, these procedures do not directly address Type II errors or test power (defined as 1 minus the probability of Type II error). In this work, we do not provide a detailed empirical or theoretical analysis of power. However, we acknowledge that studying power across different signal-to-noise and sparsity regimes (either empirically via simulations or through asymptotic approximations) would be a valuable direction for future research.
>
> To conclude the discussion on miss detection, it can be noted that the Z-test used in our setting is a likelihood ratio test. By the Neyman–Pearson lemma, such tests are known to be uniformly most powerful for simple hypotheses (see for instance chapter 3 of [9]). While our context may go beyond the classical setting of the lemma, this connection supports the intuition that our procedure is likely to exhibit reasonable power under appropriate conditions.
>
>
> > In addition to the baseline LAD method used for comparison, there appear to be other change-detection approaches, such as the statistics used in Ref [43] and related works; comparison with these methods—both empirically and conceptually—would strengthen the paper from a change-detection perspective.
>
> We thank the reviewer for raising this point. In our change detection experiment, we focused on comparing with the most relevant offline spectral method, namely the Laplacian Anomaly Detection (LAD) method, given its conceptual alignment with our approach. In contrast, Ref [43] proposes an online technique based on an extension of the CUSUM statistic, which operates under a different setting and objective. In the revised related work section, we will articulate more clearly the conceptual differences between our method and [43].
>
> However, in response to this question and reviewer HVGn’s suggestion to strengthen the change-point detection evaluation, we will add comparisons with existing methods, focusing on those with publicly available implementations to ensure fairness and reproducibility. Specifically, we will include a comparison with TensorSplat [10], a widely used approach based on PARAFAC decomposition of the temporal adjacency tensor.
>
> > Line 240 typo: should be "conservative."
>
> We thank the reviewer for pointing out this typo and corrected it in the revised version.
>
> ***References***
>
> [1] Jones, A., & Rubin-Delanchy, P. (2020). The multilayer random dot product graph. *arXiv preprint arXiv:2007.10455*.
>
> [2] Zhang, X., Xue, S., & Zhu, J. (2020, November). A flexible latent space model for multilayer networks. In *International Conference on Machine Learning* (pp. 11288–11297). PMLR.
>
> [3] Arroyo, J., Athreya, A., Cape, J., Chen, G., Priebe, C. E., & Vogelstein, J. T. (2021). Inference for multiple heterogeneous networks with a common invariant subspace. *Journal of Machine Learning Research*, *22*(142), 1–49.
>
> [4] Gallagher, I., Jones, A., & Rubin-Delanchy, P. (2021). Spectral embedding for dynamic networks with stability guarantees. *Advances in Neural Information Processing Systems*, *34*, 10158–10170.
>
> [5] Jing, B. Y., Li, T., Lyu, Z., & Xia, D. (2021). Community detection on mixture multilayer networks via regularized tensor decomposition. *The Annals of Statistics*, *49*(6), 3181–3205.
>
> [6] Agterberg, J., Lubberts, Z., & Arroyo, J. (2025). Joint spectral clustering in multilayer degree-corrected stochastic blockmodels. *Journal of the American Statistical Association*, (just-accepted), 1–23.
>
> [7] Dunn, O. J. (1961). Multiple Comparisons Among Means. Journal of the American Statistical Association, 56(293), 52–64.
>
> [8] Benjamini, Yoav, and Yosef Hochberg. “Controlling the False Discovery Rate: A Practical and Powerful Approach to Multiple Testing.” Journal of the Royal Statistical Society: Series B (Methodological) 57, no. 1 (1995): 289–300.
>
> [9] Lehmann, E. L., and Joseph P. Romano. Testing Statistical Hypotheses. 3rd ed. Springer Texts in Statistics. Springer, 2005.
>
> [10] Koutra, D., Papalexakis, E. E., & Faloutsos, C. (2012). TensorSplat: Spotting Latent Anomalies in Time. Proceedings of the 16th Panhellenic Conference on Informatics, 144–149.

---

> > ### Comment · Reviewer_AekB · 2025-08-07
> >
> > I want to thank the authors for the careful response, which addresses some of earlier concerns. I have adjusted my score accordingly. I have one additional suggestion: the current analysis of Type-I and Type-II errors primarily reflects a multiple hypothesis testing perspective. An alternative viewpoint is through the lens of change-point detection, where detection accuracy is typically evaluated using metrics such as the average detection delay (in online settings) or the absolute error between the estimated and true change-point (in offline settings). It may be worthwhile, perhaps as future work, to extend the analysis to incorporate such metrics from the change-point detection literature.

---

> > > ### Author Response · Authors · 2025-08-07
> > >
> > > We thank Reviewer AekB once again for their valuable feedback. We agree that evaluating change point detection performance in terms of temporal accuracy is an important direction for future research, and we will take this aspect into account in our future work on anomaly detection in computer networks.

---

### Official Review · Reviewer_n6aj · 2025-07-03

**Clarity:** 3
**Significance:** 3
**Originality:** 3
**Rating:** 5
**Confidence:** 4

**Summary:**

For the problem of detecting structural changes in dynamic network data, the authors propose a multi-resolution framework called ANIE (Adaptive Network Intensity Estimation). Instead of binning over fixed time intervals and comparing changes in graph structure, ANIE models interactions as Poisson processes by estimating a low-dimension subspace for node behavior, and then quantifying change in node behavior using empirical affinity coefficients. This allows for detection of both rapid and gradual changes, without static time binning.

The first stage of ANIE uses truncated SVD over empirical coefficients to estimate a global subspace matrix Subsequently, these are denoised through statistical thresholding. The empirical affinity coefficient are computed by combining node pair coefficients weighted by latent factor affinity.

The authors conduct empirical analysis using synthetically constructed networks, following Erdos-Renyi approach and stochastic block models and on real world UCI messages data. The proposed approach effectively captures perturbations of underlying intensity across various temporal resolutions, demonstrating robustness to noise.

**Questions:**

1. Elaborate in more detail on statistical thresholding for denosing.

2. More empirical analysis on an expanded set of graph network datasets.

3. The section on computational efficiency is relevant but short and vague. Please refine "For a number of nodes below a few thousands, our method runs in a few seconds on a standard laptop." to provide more specific details of configurations and results.

**Ethical Concerns:**

["NO or VERY MINOR ethics concerns only"]

**Limitations:**

Yes

**Paper Formatting Concerns:**

No concerns

**Quality:**

3

**Strengths And Weaknesses:**

The work is well written, the methodology is sound and the problem is well motivated. The experiments are conducted on a mix of synthetic and real world datasets and support the claims. The experiments are somewhat limited and could be made stronger with more network datasets.

---

> ### Author Rebuttal · Authors · 2025-07-30
>
> We thank reviewer n6aj for their time and thoughtful evaluation of our work. We are pleased that the reviewer found the manuscript well written, the motivation clear, and the problem well posed. Below, we provide a point-by-point response to the reviewer’s questions.
>
> > Elaborate in more detail on statistical thresholding for denoising.
>
> In the context of wavelet analysis, statistical thresholding is a technique used to estimate a clean signal $\\hat{x}(t)$ from noisy observations $x(t)$. It generally involves three steps:
>
> 1. *Analysis*: An isometric transformation  $ T $ is applied to the signal to obtain a new representation  $ w $ (e.g., via wavelet decomposition,  $ x \\mapsto w$), such that the underlying signal in  $ x $ is supported by only a few components of  $ w$. Examples of such transformations include the Fourier transform, wavelet transform, splinets, etc. A good transformation is one in which the signal present in  $ x $ is captured by only a small number of non-zero components in  $ w$. Book [1] discusses these sparse representations (in Chapter 1) and their applications to denoising (in Chapter 11).
>
> 2. *Statistical Thresholding*: Non-significant entries of $w$ are determined using statistical testing (hence the term statistical thresholding) where the null hypothesis is that they are 0. This results in a thresholded set of coefficients $\\hat{w}$ which only retains the signal components. A common approach is to assume that, under the null hypothesis $w^b = 0$, the coefficients $w^b$ follow a zero-centered Gaussian distribution with variance estimated from the data. This yields a set of $p$-values, which are compared to a confidence threshold. Since each component in $w$ corresponds to a hypothesis, multiple testing correction is typically applied to control the Family-Wise Error Rate (FWER) or the False Discovery Rate (FDR). We refer the reviewer to the seminal [2] for an example of statistical thresholding in this context.
>
> 3. *Synthesis*: Finally, the inverse transformation $T^{-1}$ is applied to obtain the denoised signal $\\hat{x}(t)$ from the denoised coefficients $\\hat{w}$.
>
> Our framework follows this three step approach, where
> 1. The analysis is done by calculating the coefficients $\\mathbb{Y}(\\phi_{jk})$, estimating $\\hat{U}$ and then doing projection to get the empirical affinity matrix $\\hat{S}$.
> 2. Thresholding is done globally: our procedure decides which of the $D\\times D\\times 2^J$ coefficients of $\\hat{S}$ are significant (i.e. rejecting the hypothesis that this coefficient is zero), and which are not.
> 3. The synthesis, or reconstruction, step converts the thresholded wavelet coefficients back into signals. A low-rank approximation is then performed by multiplying the result on the left and right by $\\hat{U}$ and $\\hat{U}^T$, respectively, which serves as a form of denoising, leading eventually to a reconstructed intensity matrix.
>
>
> > More empirical analysis on an expanded set of graph network datasets.
>
> To demonstrate the applicability of our method to a broader range of real-world datasets, we will include in the appendix of the revised version additional experiments on both the Enron email dataset and the London Santander Bike dataset (a dynamic graph dataset previously studied for instance in [3]), using data from May 1 to May 8, 2017.
>
> *Details of the additional experiments*
>
> For both datasets, we plot two discrete wavelet scaleograms which we here refer to as the naive and reconstructed scaleograms, by computing for each level $j$, the Frobenius norm of either the naive coefficient estimates $\lVert\mathbb{Y}(\phi_{jk})\rVert_F$ or the empirical affinity coefficients $\lVert\hat{S}(\phi_{jk})\rVert_F$. It can be shown that the latter corresponds to the Frobenius norm of a low-rank reconstruction of the naive coefficients. Specifically, using the orthonormality of $\hat{U}$, we have
> $
> \lVert\hat{S}(\phi_{jk})\rVert_F = \lVert\hat{U} \hat{U}^T \mathbb{Y}(\phi_{jk}) \hat{U} \hat{U}^T\rVert_F.
> $
> In both cases, we typically observe that the wavelet power of the reconstructed affinities between latent factors is more concentrated in low-frequency bands, while the naive per-edge estimates $\lVert\mathbb{Y}(\phi_{jk})\rVert_F$ exhibit more energy in the high-frequency bands.
>
>
>
> For the Enron dataset, we find that both the naive and reconstructed scaleograms capture changes in 2001, which marked the buildup to the company’s bankruptcy. These changes are concentrated in a specific frequency band, illustrating the ability of our method to identify the time scale at which structural shifts occur. Additionally, we observe that, at the edge level, events following 2001 are associated with changes across a wider range of frequency bands. In contrast, the reconstructed scaleogram shows less variation during this later period, suggesting that many of the post-2001 edge-level fluctuations do not correspond to substantial structural changes in the network, or at least not to the same extent as during the major 2001 events.
>
> In the London Bike dataset, plotting the naive and reconstructed wavelet scaleograms allows us to identify periods and time scales during which significant structural changes occur, and to contrast these with changes that are merely due to fluctuations in exchange intensity between individual pairs of bike stations. Moreover, the $\\hat{U}$ estimate obtained using ANIE enables us to visualize the bike stations in a low-dimensional space using a t-SNE plot, where stations that connect to similar neighborhoods at similar times appear close together. Coloring the bike stations by London borough reveals that stations from some boroughs, such as Tower Hamlets or Newham, tend to cluster tightly. In contrast, stations from boroughs like Westminster, Kensington and Chelsea, or Hammersmith and Fulham are more dispersed, indicating greater diversity in their patterns of use.
>
> > The section on computational efficiency is relevant but short and vague. Please refine "For a number of nodes below a few thousands, our method runs in a few seconds on a standard laptop." to provide more specific details of configurations and results.
>
>
> We would like to draw attention to the timed experiment included in the appendix, where we report both the specifications of the machine used to run the experiments, along with the computation times of our method for varying numbers of nodes and different values of the hyperparameter $J$. However, we acknowledge that the sentence referenced by the reviewer should be made more specific. In the main body of the revised version, we will clarify the configuration used, report specific timing results, and include an explicit reference to the relevant experiment in the appendix.
>
>
> ***References***
>
> [1] Mallat, S. G. A Wavelet Tour of Signal Processing: The Sparse Way. 3rd ed. Elsevier/Academic Press, 2009.
>
> [2] Donoho, David L., Iain M. Johnstone, Gérard Kerkyacharian, and Dominique Picard. “Density Estimation by Wavelet Thresholding.” The Annals of Statistics 24, no. 2 (1996): 508–39.
>
> [3] Sanna Passino, F., Che, Y., & Cardoso Correia Perello, C. (2024). Graph-based mutually exciting point processes for modelling event times in docked bike-sharing systems. Stat, 13(1), e660.

---

### Note · Authors · 2025-08-13

Thanks to the reviewers’ constructive feedback, the rebuttal period was highly productive. We truly appreciate the recognition of our work’s strengths, including: *“The work is well written, the methodology is sound, and the problem is well motivated”*, *“ANIE \[…] demonstrates good empirical performance across diverse datasets”*, *“the main strength of the method is the rigorous model behind the approach”*, and *“the flexibility of the model \[…] and the theoretical analysis, \[…] are all strengths of the proposed approach”*.

The discussion led to several improvements, summarized below:

* **Strengthened connection to related work** – We clarified how our method relates to prior approaches (TensorSplat, COSIE, IPP, and the method by Yu et al., ref. \[43]).
* **Refined presentation of the results** – We improved the visualizations and provided a more intuitive explanation of the change detection results.
* **Expanded empirical evaluation** – We added two new datasets (Enron and London Santander Bike) and included a direct comparison with the TensorSplat method in addition to the LAD method for change detection.
* **Theoretical considerations** – Two key theoretical insights emerged from the discussion: (i) understanding the power of the proposed test and its connection to change detection is an important open question, and (ii) we clarified the nature of the estimated subspace $\hat{U}$, distinguishing it from that obtained in traditional spectral clustering. Notably, these insights help understand better the current limitations of our method.

Several promising avenues for future work emerged from the discussion, such as:

* Studying the theoretical and empirical power of the multiple testing scheme, as mentioned above.
* Enhancing change detection evaluation using additional metrics such as Adjusted F1 or detection delay.
* Extending the method to support online, real-time change detection.

We believe these clarifications, additions, and new results have made the paper stronger, and we are once again grateful to the reviewers for helping us improve it.

---

### Decision · Program_Chairs · 2025-09-17

**Decision:**

Accept (poster)

**Comment:**

This paper proposes a multi-resolution approach for network structure change detection and estimation. Instead of using a fixed time resolution, the proposed method uses a Poisson process model for the network structure as a function of $t$, and is therefore able to adapt to changes in network structure without pre-defining a timescale. A partial theoretical analysis is provided for the subspace estimation performance and asymptotic behavior of the algorithm.

The initial reviews of the paper is quite positive, and the discussion has resulted in a number of useful suggestions for improving the paper. Overall, the paper is above the acceptance threshold if the authors can incorporate these comments into their camera ready version.